# Droplet collection efficiencies inferred from satellite retrievals constrain effective radiative forcing of aerosol-cloud interactions

Charlotte M. Beall[1], Po-Lun Ma[1], Matthew W. Christensen[1], Johannes Mülmenstädt[1], Adam Varble[1], Kentaroh Suzuki[2], Takuro Michibata[3]

[1]Atmospheric Sciences and Global Change Division, Pacific Northwest National Laboratory, Richland, WA, 99354, U.S.A.

[2]Atmosphere and Ocean Research Institute, University of Tokyo, Chiba, 277-8568, Japan

[3]Department of Earth Science, Okayama University, Okayama, 700-8530, Japan

*Correspondence to:* Charlotte M. Beall; charlotte.beall@pnnl.gov

**Abstract.** Process-oriented observational constraints for the anthropogenic effective radiative forcing due to aerosol-cloud-interactions (ERFaci) are highly desirable because the uncertainty associated with ERFaci poses a significant challenge to climate prediction. The Contoured Frequency by Optical Depth Diagrams (CFODD) analysis supports evaluation of model representation of cloud liquid to rain conversion processes because the slope of a CFODD, generated from joint MODerate Resolution Imaging Spectroradiometer (MODIS)-CloudSat cloud retrievals, provides an estimate of cloud droplet collection efficiency in single-layer warm liquid clouds. Here we present an updated CFODD analysis as an observational constraint for the ERFaci due to warm rain processes and apply it to the U.S. Department of Energy's Energy Exascale Earth System Model version 2 (E3SMv2). A series of sensitivity experiments shows that E3SMv2 droplet collection efficiencies and ERFaci are highly sensitive to autoconversion, the rate of mass transfer from cloud liquid to rain, yielding a strong correlation between the CFODD slope and the shortwave component of ERFaci (ERFaci$_{SW}$; Pearson's R = -0.91). We estimate ERFaci$_{SW}$, constrained by MODIS-CloudSat, by calculating the intercept of the linear association between the ERFaci$_{SW}$ and the CFODD slopes, using the MODIS-CloudSat CFODD slope as a reference. When E3SMv2's CFODD slope is constrained to agree with the A-Train retrievals, ERFaci$_{SW}$ is reduced by $14 \pm 6\%$ in magnitude, indicating that correcting bias in the ERFaci$_{SW}$ due to autoconversion would bring E3SMv2's total ERFaci ($-1.50$ W m$^{-2}$) into better agreement with the IPCC AR6 'very likely' range for ERFaci ($-1.0 \pm 0.7$ W m$^{-2}$).

## 1 Introduction

Single-layer, low-level marine warm clouds cover 25% of the ocean surface (Charlson et al., 1987) and exert a strong cooling effect on climate due to their reflectivity (Hartmann et al., 1992; Hartmann and Short, 1980; Ramanathan et

al., 1989). Aerosols modulate multiple radiative properties of low warm clouds, including droplet number
concentration ($N_d$), liquid water path (LWP), geometric , cloud fraction, and lifetime, and their net impact on the cloud
radiative forcing  is the most uncertain component of the climate system  (e.g., Stevens and Feingold, 2009;
Christensen et al., 2020; Glassmeier et al., 2021). Though aerosols also exert a significant influence on ice and mixed-
phase clouds, aerosol-cloud interactions (ACI) make their largest contribution to global radiative forcing via liquid
water clouds (Bellouin et al., 2020).
In marine warm cloud regimes, an increase in aerosol concentrations typically leads to increasing $N_d$. Given constant
condensed water content, enhanced aerosol concentrations increase cloud albedo due to higher concentrations of
smaller cloud droplets through the so-called "Twomey effect" (Twomey, 1974). However, the cooling effect of
increased $N_d$ can be offset or enhanced by competing aerosol-mediated cloud properties such as cloud fraction and
LWP. For example, increased numbers of smaller droplets can diminish cloud fraction by reducing cloud droplet
sedimentation (Bretherton et al., 2007) and increasing cloud-top evaporation and dry air entrainment (Wang et al.,
2003). On the other hand, aerosols can also increase cloud fraction and vertical extent by suppressing precipitation
(Albrecht, 1989; Pincus and Baker, 1994). Christensen et al. (2020) demonstrated that the impact of aerosol on low-
level cloud areal coverage depends on the stability of the atmosphere: in thermodynamically stable lower tropospheric
conditions, increased aerosol results in increased cloud fraction, lifetime and $N_d$, whereas in unstable conditions,
entrainment and evaporation offset Twomey effects, resulting in relatively smaller changes to cloud radiative
properties.
Earth Systems Models (ESMs) are relied upon for estimating the global Effective Radiative Forcing of Aerosol-Cloud
Interactions (ERFaci) due to the dearth of observations from the pre-industrial era. Yet ESM estimates are challenged
by the lack of observational constraints on ERFaci and the cloud processes that modulate ERFaci, which must be
parameterized due to the computational expense of explicitly resolving them. Mülmenstädt et al. (2020)  proposed a
renewed focus on process-oriented observational constraints as a solution to "equifinality", whereby differing
representations of cloud processes can reproduce observed state variables such as LWP and cloud fraction. The
problem of equifinality renders many global long-term observations of state variables useless for constraining ERFaci
on their own. Mülmenstädt et al. (2020)  argues that constraints based on cloud process observations are thus highly
desirable as an alternative approach to state variable-based constraints because mitigating bias in a cloud process

representation will improve estimates of the response of the process to aerosols. Process-oriented constraints on ERFaci are useful for quantifying the sensitivity of ERFaci to a specific process or constraining the component of ERFaci that is affected by a process,  rather than for constraining ERFaci overall (Mülmenstädt and Feingold, 2018). Recent examples of process-based diagnostics include the Earth System Model Aerosol-Cloud Diagnostics Package (ESMAC Diags) (Tang et al., 2022; Tang et al., 2023), which supports evaluation of aerosol activation processes, and Varble et al. (2023) which demonstrated multiple model-observations comparison approaches that target processes affecting cloud albedo susceptibility using geostationary satellite data and surface-based observations. Christensen et al. (2023) applied ground-based measurements, satellite retrievals and meteorological reanalysis products in a Lagrangian framework to evaluate multiple aerosol-cloud processes in E3SM, including cloud condensation nuclei deposition via precipitation and the temporal response in $N_d$ to aerosol perturbations.

In response to the demand for process-oriented constraints on warm liquid cloud processes, we present a constraint on the shortwave component of ERFaci (ERFaci$_{SW}$) due to autoconversion, a parameterization representing the transfer of liquid mass and number from the cloud to rain category, based on satellite cloud retrievals. For the past 12 years, prior studies have applied the Contoured Frequency by Optical Depth Diagrams (CFODD) analysis (Nakajima et al. 2010; Suzuki et al. 2010) to evaluate model representation of warm rain processes because the slopes of CFODDs, generated from spaceborne radar reflectivity profiles (CloudSat) (e.g. Marchand et al., 2008) and cloud property retrievals from the Moderate Resolution Imaging Spectroradiometer (MODIS) (e.g. Platnick et al., 2017), provide an estimate of cloud droplet collection efficiency in warm liquid clouds (Suzuki et al. 2010). Here we demonstrate how an updated CFODD analysis can be applied to constrain ERFaci due to autoconversion usingthe U.S. Department of Energy's Energy Exascale Earth System Model version 2 (E3SMv2) and the relationship between CFODD slopes and ERFaci$_{SW}$ in SLWCs.

To support the application of CFODD analysis as a constraint on ERFaci$_{SW}$, we modified the Warm Rain Diagnostics (WRDs) subroutine (Michibata et al. 2019) that was recently implemented in the Cloud Feedback Model Intercomparison Project (CFMIP) Observations Simulator Package (COSPv2.0), a software package that supports climate model evaluation against satellite observations (Michibata et al., 2019; Swales et al., 2018). The WRDs support evaluation of model warm rain processes in single-layer warm liquid clouds (SLWCs) based on joint statistics from MODIS and CloudSat. The first diagnostic provides the fractional occurrence of SLWCs, classified as non-

precipitating, drizzling, or raining clouds based on CloudSat column maximum radar reflectivity. The second
diagnostic is the CFODD, which is the probability density function (PDF) of radar reflectivity as a function of in-
cloud optical depth (ICOD), where ICOD is the optical depth integrated from the cloud top downward to each vertical
layer and represents an in-cloud vertical coordinate (Nakajima et al., 2010; Suzuki et al., 2010). The CFODD shows
how vertical cloud microphysical structures transition from non-precipitating to precipitating as a function of cloud-
top effective radius ($R_e$), and the slope of reflectivity change with ICOD provides an estimate of droplet collection
efficiency factor (Suzuki et al., 2010). Previous studies have used CFODDs to demonstrate that pollution decreases
droplet collection efficiency, suppressing rainfall near the cloud base (Mangla et al., 2020; Michibata et al., 2014;
Suzuki et al., 2013), and to evaluate model cloud liquid to rain conversion processes against satellite observations
(Suzuki et al., 2015; Jing et al. 2019; Michibata and Suzuki, 2020). Takahashi et al. (2021) proposed an updated
CFODD analysis in which $R_e$ thresholds are defined by quartile distributions of SLWC samples rather than the
traditional CFODD $R_e$ thresholds to focus evaluation on warm rain process representation rather than the bias in $R_e$
distribution. Modifications to the WRDs in the present study include additional diagnostics that provide SLWC
sampling statistics to illuminate how sample size affects CFODD results, the implementation of a CloudSat ground-
clutter mask in the simulated WRDs and updates to SLWC selection criteria for better consistency between
observations and satellite simulators.  The updated CFODD analysis is demonstrated here as a constraint on the
component of ERFaci$_{SW}$ that is affected by droplet collection efficiency due to autoconversion.
**2 Warm Rain Diagnostics Overview**
The WRDs and their implementation in COSPv2.0 were described in Michibata et al. (2019). The WRDs are designed
to run online with the host model, accumulating time step statistics on warm rain cloud processes for subcolumns to
mitigate the risk of data-processing bottlenecks associated with outputting large data volumes. COSPv2.0 generates
ensembles of stochastic subcolumns from model gridbox-mean variables to emulate model subgrid variability and to
resolve discrepancies in spatial resolution between observations and the model grid (Swales et al., 2018).
To generate observational reference data for model evaluation, Michibata et al. (2019) used the MODIS and CloudSat
products 2B-TAU R04 (Polonsky, 2008) and 2B-GEOPROF R04 (Mace et al., 2007; Marchand et al., 2008),
respectively, for SLWC detection between June 2006 and April 2011. The  SLWC detection are described in
Supplement Table S1 and include CloudSat reflectivity $\geq$ -30 dBZ, MODIS liquid COT > 0.3, and cloud top
temperature $\geq$ 273 K. Model-simulated SLWCs are detected using COSPv2.0 simulated CloudSat reflectivity and
multiple MODIS cloud properties, including ice water path (IWP), liquid water path (LWP), cloud-top effective radius
($R_e$), and cloud optical thickness (COT) (Table S1). For the SLWC fractional occurrence (frequency) diagnostic,
SLWCs are binned by precipitation intensity according to the maximum column CloudSat reflectivity ($Z_{max}$), where
non-precipitating, drizzling and raining SLWCs correspond to $Z_{max} < -15 \, dBZ_e$, $-15 \, dBZ_e \leq Z_{max} < 0 \, dBZ_e$,
and $Z_{max} \geq 0 \, dBZ_e$ , respectively. The SLWC fractional occurrence diagnostic features frequency of each
precipitation type relative to the total SLWC population.
To support evaluation of liquid cloud collection efficiencies and cloud to rain transition processes, CFODDs are
constructed from the PDFs of CloudSat reflectivity profiles binned by ICOD. ICOD ($\tau_d$) is parameterized as a function
of MODIS COT ($\tau_c$) by invoking the adiabatic condensation growth model to vertically slice the column COT into
each layer (Suzuki et al., 2010). The relationship between $\tau_d$ and $\tau_c$ is as follows:
$$\tau_d(h) = \tau_c \left\{ 1 - \left( \frac{h}{H} \right)^{5/3} \right\} \tag{1}$$
where $h$ is height and $H$ is the geometric height of the cloud. The detailed derivation of the ICOD coordinate is
provided in Suzuki et al. (2010). The slope of the resulting 2D-PDF diagnostic is modulated by droplet collection
efficiency, with steeper slope implying higher efficiency. The CFODD shows where, with ICOD on the y-axis as a
vertical coordinate, the droplet collection efficiency increases, and where the transition from non-precipitating to
drizzling and raining occurs, using the radar reflectivity as a proxy for the precipitation rate as described above (e.g.,
Muhlbaeuer et al., 2014). CFODDs are also typically binned by $R_e$ to reveal how droplet collection efficiency changes
with droplet size (Suzuki et al., 2010; Takahashi et al., 2021; Jing et al., 2017).
In this study, CFODD slopes are estimated using RANdom SAmple Consensus (RANSAC) robust linear regression
(Fischler et al., 1987). RANSAC was chosen for performing linear regression due to the right-skewed distribution of
CFODD datasets. The regression was applied to the MODIS-CloudSat profiles and E3SMv2 output at $4 \leq$ ICOD $\leq 20$
and $Z < 20$ dBZ. For E3SMv2 output, the regression was applied to approximated source CloudSat reflectivity and
ICOD data that was estimated from time-mean CFODD frequencies. The reflectivity and ICOD thresholds were were
chosen to reduce the effect of the Mie scattering regime where the radar reflectivity can be saturated and to restrict
analysis to profiles where the uncertainty of MODIS COT retrievals is lower as error can be higher in optically thin
liquid clouds (e.g., COT < 4) (Platnick et al., 2017). The uncertainty in the RANSAC slope calculation is estimated
by "bootstrapping", repeatedly performing RANSAC regressions on 1000 random subsamples of 80% the CFODD
dataset to generate a distribution of slope estimates. The 1-sigma error and 95% confidence intervals were calculated
from this distribution. The residual threshold applied for RANSAC outlier detection was 0.1 and $0.5\times$ median absolute
error (MAE) for MODIS-CloudSat and E3SMv2, respectively. Data points with MAE exceeding the residual threshold
are excluded from the linear regression in RANSAC.
**2.1 E3SMv2**
Several updates to the WRDs are described in Sect. 2.2, the impacts of which are demonstrated through an application
of the updated WRDs to the U.S. Department of Energy's Energy Exascale Earth System Model v2 (E3SMv2). The
atmosphere component of the model, E3SM Atmosphere Model v2 (EAMv2), is described in detail in Golaz et al.
(2022). Like its predecessor EAMv1, EAMv2 predicts stratiform and shallow cumulus cloud macrophysics through
the Cloud Layers Unified by Binormals (CLUBB) parameterization, which unifies the treatment of planetary boundary
layer turbulence, shallow convection, and cloud macrophysics through a higher-order turbulence closure scheme
(Bogenschutz et al., 2013; J. C. Golaz et al., 2002; Larson, 2017; Larson & Golaz, 2005). CLUBB diagnoses cloud
fraction and cloud liquid water from a joint double-Gaussian PDF. Ice and liquid cloud fractions in CLUBB are
analytically diagnosed by integrating saturated proportions of the joint PDF (Guo et al. 2015).
Cloud microphysics is represented with the "Morrison and Gettelman version 2" (MG2) scheme (Gettelman and
Morrison, 2015). MG2 prognoses the mass mixing ratios and number concentrations of cloud liquid, ice and
precipitation hydrometeors. The coupled MG2 cloud microphysics and CLUBB higher-order turbulence
parametrization explicitly provides values for hydrometer mass and number mixing ratios as well as cloud fraction.
Deep convection is represented by the Zhang and McFarlane (1995) (ZM) scheme. As convective cloud fraction is
not parameterized in the mass-flux based ZM scheme, it is diagnosed from the cloud mass flux for cloud radiation
calculation (Hack et al., 1993). The total cloud fraction in EAMv2 combines CLUBB, deep convective cloud fractions
and ice cloud fraction following (Park et al., 2014). The four-mode version of the Modal Aerosol Module (MAM4) is
used to predict aerosol properties and processes (Liu et al., 2012, 2016; H. Wang et al., 2020).
EAMv2 runs on 72 vertical atmospheric levels with a top at 0.1h Pa (Rasch et al., 2019; Xie et al., 2018). However,
distinct from its predecessor EAMv1, EAMv2 has two separate parameterized physics and dynamics grids (Hannah
et al., 2021), with average horizontal grid spacings of ~165 km and ~110 km, respectively.
A six-year E3SMv2 simulation with transient, present-day forcing was run between 2006 and 2011 with online
COSPv2.0 for comparison with A-Train observations of SLWCs, allowing one additional year (2005) for model spin-
up. To facilitate comparison with observations, large-scale winds were constrained via the "nudging" technique (Lin
et al., 2016; Ma et al., 2014; Zhang et al., 2014), in which horizontal and vertical winds are relaxed toward the Modern
Era-Retrospective Analysis for Research and Applications, Version 2 (MERRA2) reanalysis data (Gelaro et al., 2017)
with a 6-hour time-scale. MERRA2 data are read in every 3 hours and linearly interpolated to model times. COSPv2.0
is called at every time step (0.5 h) and run with 10 subcolumns. We observed little change in CFODD results for
increased numbers of subcolumns of 20 to 50.
**2.2 COSPv2.0**
Cloud-observing instrument simulators support evaluation of model cloud representation by translating gridbox-mean
model variables (e.g., cloud fraction, hydrometeor mass mixing ratio, precipitation) into quantities that are measured
by a cloud sensor (e.g., reflectivity). COSPv2.0 includes multiple cloud-observing satellite simulators and has been
used extensively to diagnose issues in model cloud representation (Cesana & Chepfer, 2012; Kay et al., 2016; Song
et al., 2018a; Y. Zhang et al., 2010). Recently, M. Zhang et al. (2022) used the COSPv2.0 CALIPSO simulator to
demonstrate that changes to the Wegener-Bergeron-Findeisen process in EAMv2 decreased an ice cloud fraction low
bias in the Arctic compared to EAMv1 but did not correct excesses of supercooled liquid.
There are known limitations to COSPv2.0 that affect its application to E3SM for cloud representation evaluation. The
subgrid distribution of cloud variables generated by COSPv2.0 does not match E3SM subgrid variability.
Hydrometeor species are distributed homogeneously across the subcolumns generated by COSPv2.0 via the
subcolumn generator SCOPS (Subcolumn Cloud Overlap Profile Sampler) (Klein and Jakob, 1999), such that the
ensemble of subcolumns reproduces the gridbox cloud fraction but not the subgrid distribution of liquid and ice within
the simulated clouds (Dewald, 2021). Song et al., (2018b) demonstrated that the default "homogeneous hydrometeor
scheme" from SCOPS results in overestimation of radar reflectivity in warm liquid clouds, thus overestimating
precipitating clouds since maximum column reflectivity is often used to distinguish precipitating clouds (as in the
WRDs). Errors in simulated satellite retrievals have also been attributed to SCOPS overlap assumptions (Hillman et
al., 2018). Such a bias from SCOPS can result in unfair observational evaluation of a host model such as E3SMv2.
Inconsistencies in microphysical assumptions between the host model and COSP pose another challenge. While many
microphysical assumptions in COSPv2.0 can be configured for agreement with E3SMv2 microphysics (MG2), some
inconsistencies remain, including gamma distribution shape parameters for hydrometeor size distributions and
hydrometeor vertical overlap assumptions (J. Wang et al., 2021). Next-generation E3SM development includes efforts
to improve agreement in the subgrid variability and microphysical assumptions involved in forward simulating
satellite retrievals. Other issues include the simplified treatment of satellite cloud detection in simulators. For example,
the CloudSat Cloud Profiling Radar (CPR) cloud mask value threshold $\geq 30$ is applied for cloud detection in the
WRDs' A-Train analysis to indicate "good" or "strong" echo with high confidence detection (see next section and
Supplement Table 1). The CPR cloud mask confidence levels consider signal-to-noise ratios, horizontal averaging,
and spatial continuity (Marchand et al., 2008), but as this cloud mask is not available in COSPv2.0, CloudSat cloud
detection is simulated by applying a reflectivity threshold $-30 \leq Z_e \leq 20$ dBZ.
The WRDs rely on COSPv2.0 simulated MODIS and CloudSat retrievals. The WRDs in COSPv2.0 work as
follows: First, COSPv2.0 takes in model atmospheric state and cloud variables including temperature, pressure,
water vapor and hydrometeor mass mixing ratios, hydrometeor $R_e$, large-scale stratiform cloud fraction, convective
cloud fraction and precipitation rate. The COSPv2.0 subcolumn generator SCOPS then produces subgrid
distributions of clouds and precipitation for better comparison with smaller scale satellite pixel measurements.
SCOPS subcolumns are homogenous, discrete samples generated such that a sufficiently large ensemble reproduces
the model column profile of bulk cloud properties (Webb et al., 2001; Swales et al., 2018). SCOPS assigns each
subcolumn a type (large-scale stratiform, convective or clear-sky) according to the host model's convective and
large-scale stratiform cloud fraction. Cloud properties such as hydrometeor mass mixing ratios and $R_e$ are distributed
homogeneously across the subcolumns by cloud type (i.e., all stratiform cloud subcolumns are assigned the same
stratiform ice and liquid mixing ratios as SCOPS only takes total convective and stratiform cloud fraction as input,
and does not consider stratiform liquid and ice cloud fraction in its default configuration. "Maximum-random"
cloud overlap is applied to subcolumns, consistent with the model parameterizations. The MODIS and CloudSat
simulators apply simplified versions of their respective retrieval algorithms to each subcolumn, emulating MODIS
retrievals of cloud properties, radar reflectivity and lidar backscatter, respectively. Gridbox-mean values are
estimated from accumulated subcolumn statistics. The WRDs take as inputs gridbox-mean simulated MODIS
retrievals of LWP, IWP, COT and $R_e$, as well as subcolumn CloudSat reflectivity profiles. The simulated MODIS
COT represents in-cloud mean, as do the other MODIS variables used in the WRDs (e.g., LWP, $R_e$ ). For example,
the MODIS liquid COT is computed by averaging the MODIS liquid COT in cloudy subcolumns across the grid-
box. In E3SMv2-COSP, the same in-cloud stratiform COT value from the E3SMv2 radiative transfer module is
distributed across all the subcolumns designated as stratiform cloud by SCOPS, as described above. These values
and cloud/clear-sky designations for each subcolumn are used as input to the MODIS simulator to calculate the in-
cloud MODIS liquid COT. Subcolumn-level SLWC reflectivity profiles are used as input to the WRDs, also with
cloud properties homogenously distributed across the subcolumns of a given classification. Thus, in E3SM-COSP,
the SLWC samples within a gridbox that have the same subcolumn classification (i.e., stratiform liquid or stratiform
rain) will have the same simulated MODIS COT and CloudSat reflectivity profile.
Deviations from the original WRDs implemented in COSPv2.0 (Michibata et al., 2019b) include the application of
the simulated CloudSat ground-clutter filter (available in COSPv2.0, but not applied to the WRDs previously) for
better comparison with CloudSat retrievals, and the elimination of the "fracout" input used in the SLWC detection
scheme from SCOPS. "Fracout" is the subcolumn-level cloud classification by vertical level from SCOPS, where each
level of each subcolumn is designated as large-scale stratiform, convective, or clear-sky. This input was removed from
the WRDs' SLWC detection algorithm because of the lack of comparable cloud-type designation in the observations
and CloudSat simulator and because "fracout" vertical cloud profiles were observed to deviate significantly from
CloudSat reflectivity profiles (i.e., fracout indicates clear-sky where CloudSat reflectivity indicates cloud, or vice
versa).
**2.3 Satellite data**
The MOD06-1KM-AUX R05 product (Platnick et al., 2017), which provides MODIS collection 6 retrievals at 1 km
resolution along the CloudSat footprint, supplied the 6 MODIS cloud retrievals required for the SLWC detection
described in Suzuki et al. (2010): LWP, IWP, $R_e$, COT, cloud top pressure and cloud layer number. Standard MODIS
products from the 2.1 µm channel were used for $R_e$, consistent with the simulated MODIS $R_e$ used in the WRDs.
Atmospheric temperature profiles were obtained from ECMWF-AUX R05 (Partain and Cronk, 2017), which includes
temperature profiles from the European Centre for Medium-Range Weather Forecast (ECMWF) model (Dee et al.,
2011) interpolated to the CloudSat footprint. 2B-GEOPROF R05 provided the CloudSat reflectivity profiles, the Cloud
Profiling Radar (CPR) cloud mask and echo top characterization at 1.8 km resolution (Marchand et al., 2008). The
detection of SLWCs and CFODD analysis in the present study follows Suzuki et al. (2010) (see Supplement Table 1
for details) with one exception: a COT threshold was decreased from 15 to 0.3; this had a substantial impact on cloud
occurrence (Figure 1; described next) and is consistent with the COT threshold implemented in the COSPv2.0 WRDs.
The decreased COT threshold also increases the weight of optically thin SLWCs, as the linear regression is applied to
the CFODD source data directly (i.e., the ICOD and reflectivity profiles).
**2.4 Autoconversion sensitivity experiments and ERFaci**
The autoconversion parameterization in E3SMv2 is a modified Khairoutdinov & Kogan (2000) scheme (hereafter,
KK2000), in which coefficients were updated in response to large uncertainties in different cloud regimes and to
improve fidelity in climate simulations. The KK2000 autoconversion scheme is $\frac{\delta q_r}{\delta t}_{auto} = A Q_c{}^{\alpha} N_d{}^{\beta}$ , where $q_r$ is
the rainwater mixing ratio, $Q_c$ is the cloud water mixing ratio, $N_d$ is the cloud droplet number concentration, and A, α
and β are the modified coefficients.
To develop a constraint on the ERFaci due to autoconversion, we performed multiple pairs of simulations featuring
preindustrial (PI) and present-day (PD) aerosol emissions. In each pair of simulations, one of the three coefficients
(A, α or β) was modified to its KK2000 value, a value reported by Wood (2005), a value from Kogan (2013)or a value
within a range bounded by the three studies. The Kogan (2013) coefficient values were derived from a large-eddy
simulation (LES) with bin resolved microphysics for cumulus clouds, whereas the focus of Wood (2005) and KK2000
was stratocumulus clouds from observational and LES perspectives, respectively. One additional experiment on the
KK2000 parameterization for the accretion rate was performed, the formulation of which is $\frac{\delta q_r}{\delta t}_{accre} =$
$F_1 F_2 67 (Q_c Q_r)^{1.15} \rho^{-1.3}$, where $Q_r$ is the rain water mixing ratio, $F_1$ represents subgrid $Q_c$ variability, $\rho$ is air density,
and $F_2$ is an accretion rate enhancement factor. $F_2$ was increased by a factor of ∼ 3 in the accretion sensitivity
experiment. $F_2$ is considered a tunable parameter in E3SM (Ma et al., 2022). The experiment details are provided in
Table 1.
Table 1. KK2000 coefficient and accretion enhancement factor values applied in 12 sensitivity experiments. Dash ("-
") indicates the coefficient value was unchanged from the default E3SMv2 parameterization (equal to the "CNTL"
simulation value).

| Name | A | $\alpha$ | $\beta$ | accre |
|---|---|---|---|---|
| CNTL | 3.05E4 | 3.19 | -1.4 | 1.75 |
| alpha01 | - | 4.22 | - | - |
| beta01 | - | - | -1.0 | |
| acoef100x | 3.05E6 | - | - | - |
| alpha02 | - | 2.47 | - | - |
| acoef0.05x | 1.35E3 | - | - | - |
| alpha03 | - | 3.00 | - | - |
| beta03 | - | - | -1.79 | - |
| beta04 | - | - | -3.01 | - |
| acoef10x | 3.05E5 | - | - | - |
| acoef5x | 1.53E5 | - | - | - |
| acoef50x | 1.53E6 | - | - | - |
| accre01 | - | - | - | 5 |


ERFaci for each pair of simulations was calculated following the Ghan (2013) method, where $ERFaci = \Delta(F_{clean} -$
$F_{clear,clean})$. $F_{clean}$ is the radiative flux at the top-of-atmosphere (TOA) neglecting the absorption and scattering of
aerosols, and $F_{clear,clean}$ is the radiative flux at the TOA neglecting both clouds and the absorption and scattering of
aerosols. The $\Delta$ indicates the PD – PI difference. While the PD-PI approach is a common strategy for estimating
ERFaci, Christensen et al. (2023) demonstrated that it may yield a different estimate than the PD approach, where
components of ERFaci (LWP adjustment, $N_d$ adjustment, cloud fraction adjustment) are estimated by regressions of
cloud properties multiplied by the anthropogenic aerosol fraction. We calculate ERFaci for SLWCs only, binned by
the MODIS $R_e$ range corresponding to the CFODD analysis.
A constraint on ERFaci$_{SW}$ was calculated from the linear regression between E3SMv2 CFODD slopes and ERFaci$_{SW}$,
using the MODIS-CloudSat CFODD slope as a reference. A 95% confidence interval for the linear fit was estimated
by bootstrapping the linear regression within the uncertainty of the CFODD slopes. CFODD slope values were
randomly sampled 1000 times within their 1-sigma error and repeatedly regressed with ERFaci$_{SW}$. The original data
(i.e., RANSAC CFODD slope values and corresponding ERFaci$_{SW}$ values) were additionally resampled with
replacement to generate a distribution of coefficients for the ordinary least squares (OLS) regression. The 95%
confidence interval for the linear fit was then calculated from the combined linear regression coefficient distributions
to reflect uncertainty from both the OLS fit and the CFODD slopes.
**3 Updates to MODIS and CloudSat SLWC analysis and reference data**
The first diagnostic in the original WRDs featured relative frequencies of SLWCs by precipitation intensity in both
the A-Train reference data and the COSPv2.0 output (e.g., Fig. 1 m-o). We have updated this diagnostic with all-sky
frequencies and by decreasing the lower MODIS COT threshold from 15 to 0.3, for consistency with the WRDs
implemented in COSPv2.0 (Fig. 1 a-l). SLWCs featured in Fig. 1 and all following figures and analyses are ocean-
only due to higher uncertainties in MODIS retrievals over land (Platnick et al., 2017).

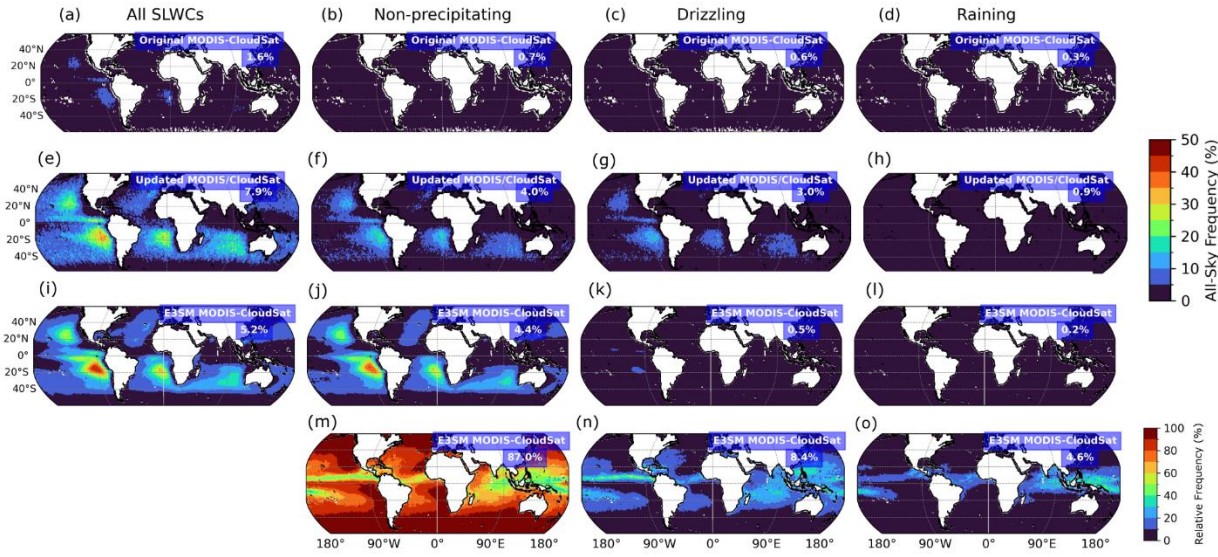

**Figure 1.** All-sky frequencies of total SLWCs June 2006 – Apr 2011, non-precipitating ($Z_{max} < -15\ dBZ_e$), drizzling
($-15\ dBZ_e \le Z_{max} < 0\ dBZ_e$) and raining ($Z_{max} \ge 0\ dBZ_e$) ocean-only SLWCs according to original reference analysis of
MODIS and CloudSat observations (Michibata et al., 2019a, 2019b) (a-d), updated reference MODIS and CloudSat analysis (e-h)
and E3SMv2-COSPv2.0 (i-l). Figures m-o show frequencies of non-precipitating, drizzling and raining SLWCs relative to the total
SLWCs simulated by E3SMv2. Values in blue boxes indicate global ocean-only grid-weighted mean frequency. SLWCs were
undersampled in original reference A-Train analysis by a factor of ~5. Compared to the original reference A-Train data, the updated
analysis demonstrates that E3SM underrepresents rather than overrepresents total SLWC frequency and that precipitating SLWCs
are underrepresented by a factor of 6 compared to observations.
Figure 1 also shows that decreasing the lower MODIS COT threshold from 15 to 0.3 in the updated A-Train analysis
(Sect. 2.3) increased total SLWC sampling by 5-fold (global ocean mean, see Sect. 2.3) compared to the original
CFODD analysis in Michibata et al. (2019a) and Michibata et al. (2019b). The increase in SLWC sampling in the
reference data affects multiple outcomes of the model evaluation in this case: E3SMv2 underrepresents, rather than
overrepresents, total SLWCs, and the SLWCs that are missing from E3SMv2 are entirely the precipitating SLWC
populations. The underrepresentation of precipitating SLWCs in E3SMv2-COSP indicates that any bias from SCOPS
towards increased precipitation in warm liquid clouds is relatively minor (Sect. 2.2; Song et al. (2018)). Not all the
differences between the original and updated reference data can be explained by the change in COT threshold,
however, as we were unable to reproduce the original CFODD data with the updated satellite products used in this
study. Fig. S1 and S2 show that increasing the lower COT threshold from 0.3 to 15 yields SLWC frequencies that are
much closer to the original reference data (+25%) than the updated reference data, but significant differences remain
in the CFODDs.
The effects of the increased SLWC sampling in the A-Train reference data also significantly affected the CFODDs
and thus the comparison between A-Train and E3SMv2 droplet collection efficiencies. Figure 2 shows CloudSat
reflectivity frequency binned by ICOD for the original A-Train reference data (Fig. 2 a-c), the updated A-Train
reference data (d-f) and E3SMv2 (j-l), and RANSAC robust linear regression slopes at $4 \leq ICOD \leq 20$. In comparisons
with various other linear regression techniques, we found that RANSAC best supported the comparison of CFODD
slopes between E3SMv2 and observations because of the right-skewed distribution of CloudSat reflectivities at $0 \leq$
$ICOD \leq 20$ in E3SMv2 CFODDs (Figs. 2 j-l). RANSAC minimizes the median absolute error (MAE) and is less
sensitive to strong outliers in the dimension of the predicted variable ($Z_e$ in this case) compared to other linear
regression techniques.
The updated A-Train CFODD distributions are significantly different than the original CFODD distributions (2D-
Kolmogorov-Smirnov test, $p \ll 0.05$). Compared to updated A-Train CFODDs, the E3SMv2 CFODDs show
decreased droplet collection efficiencies and an increased range of reflectivities near the cloud top in all size bins,
indicating that regardless of $R_e$, SLWCs are drizzling and raining near the cloud top with significantly higher frequency
than SLWCs in observations but have decreased collection efficiency below cloud top compared to MODIS-CloudSat.

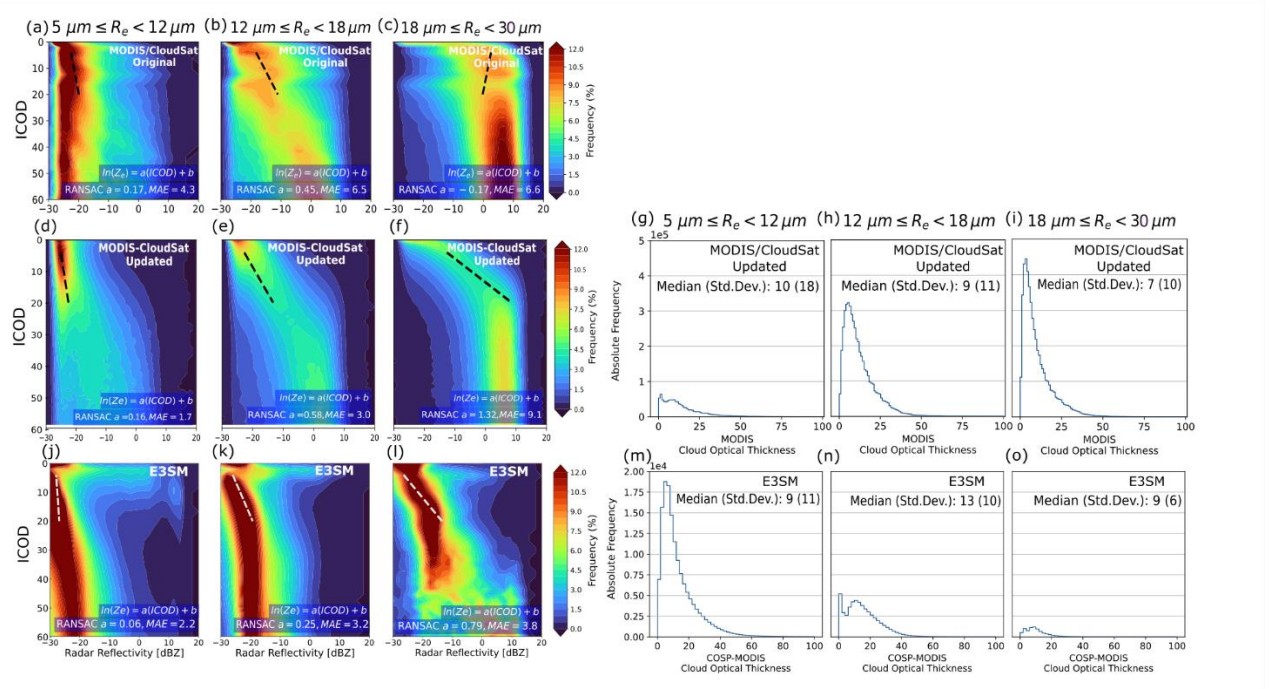


**Figure 2.** Contoured frequency by optical depth diagrams (CFODDs) for SLWCs June 2006 – April 2011 binned by MODIS cloud

top effective radius ($R_e$) from original reference MODIS-CloudSat observations analysis (a-c), updated reference MODIS-CloudSat

observations analysis (d-f), and E3SMv2 (j-l). Random Sample Consensus (RANSAC) linear regressions were applied to the

CFODD at $4 \le ICOD \le 20$ to estimate droplet collection efficiencies. RANSAC slopes and Median Absolute Error (MAE) values

are shown in blue boxes. Droplet collection efficiencies increase with MODIS $R_e$ as expected, except for the largest $R_e$ size bin in

the original reference data (Fig. s2c). Figs. g-i and m-o show absolute frequencies of SLWCs by MODIS COT, demonstrating that

E3SMv2 overrepresents SLWCs with small $R_e$ relative to medium and large $R_e$, compared to observations.

The high reflectivities near the cloud top are pronounced in the subset of E3SMv2 SLWCs with $4 < $ MODIS COT $<$
20 (Fig. S3), indicating that the high reflectivity at low ICOD in Figs. 2 (j-l) is not just a product of a subset of
precipitating, optically thin SLWCs, but that layers near the cloud top in deeper SLWCs are also precipitating. also
contribute. The reflectivity profiles used to generate the CFODD come from the CloudSat simulator, which was not
modified for this study. Examples of simulated CloudSat reflectivity profiles in SLWCs with $Z_e > 0$ dBZ near cloud
top are shown in Fig. S4.    The source of this issue and its implications for E3SMv2 representation of liquid cloud
properties warrant further investigation that is beyond the scope of the present study.
Figure 2 shows absolute frequencies of SLWCs binned by MODIS COT in each CFODD $R_e$ bin for the updated A-
Train analysis (Fig. 2 g-i) and E3SMv2 only (Fig. 2 m-o). Note, this information was unavailable in the original
reference data (Michibata et al., 2019a). Compared to COT distributions in the updated A-Train analysis, E3SMv2
shows decreasing SLWC frequency with $R_e$ and an underrepresentation of SLWCs with large $R_e$, which aligns with
the underrepresentation of precipitating SLWCs in Fig. 1. Fig. 2o also shows that few SLWCs with large $R_e$ have a
COT > 20, indicating that the CFODD reflectivity profile in Fig. 2l at ICOD > 20 is comprised of few samples. The
SLWC COT PDFs have been implemented in the WRDs to support the interpretation of CFODD statistics.
**4 Results and Discussion**
**4.1 CFODD analysis to constrain ERFaci due to warm rain processes**
To demonstrate the potential of the CFODD analysis described above for constraining $ERFaci_{SW}$ due to warm rain
processes, we performed 12 experiments featuring variations of E3SMv2's autoconversion and accretion
parameterizations, computing $ERFaci_{SW}$ for the SLWC samples represented in each CFODD and the corresponding
$R_e$ bin (hereafter, "$ERFaci_{SW\_SLWCs}$") following Ghan (2013; see Sect. 2.4). In each experiment, a single coefficient of
either the KK2000 autoconversion or accretion parameterization was perturbed, each of which is treated as a tunable
parameter in E3SMv2. The uncertain KK2000 coefficients, coupled with parameterization simplifications (e.g., bulk
moments and assumed droplet size distributions), result in uncertainties and biases in the model representation of
raindrop formation and growth. The experiments are described in Table 1, and the CFODDs for each experiment are
shown in Fig. S5.
Figure 3 shows a strong negative correlation between E3SMv2 $ERFaci_{SW\_SLWCs}$ with "small" or "medium" $R_e$ (i.e., 5
$\leq R_e < 18$ µm) and the corresponding combined $5 \leq R_e < 18$ µm CFODD slope ( Pearson's R = -0.91). SLWCs with
large $R_e$ ($18 \leq R_e < 30$ µm) were excluded from the analysis in Fig. 3 because this population represents a negligible
fraction of total SLWCs in E3SMv2 (see Fig. S6), resulting in poor sampling statistics and larger regression
uncertainties. As CFODD slopes represent an estimate of droplet collection efficiency, Fig. 3 demonstrates that
ERFaci$_{SW}$ strengthens (increases in magnitude) with increasing droplet collection efficiency in E3SMv2 SLWCs with
R$_e$ between 5 and 18 µm. One possible physical explanation for the relationship between autoconversion, droplet
collection efficiency, and ERfaci$_{SW}$ is that increased autoconversion rates increase the susceptibility of clouds to
precipitation suppression by aerosols. For a given optical depth, SLWCs with lower LWP and/or higher N$_d$ will
precipitate more when the autoconversion rate is increased. A larger population of precipitating SLWCs results in
increased susceptibility to precipitation suppression by aerosols overall. When aerosols suppress precipitation (e.g.,
Suzuki et al., 2013), LWP and/or cloud fraction may be enhanced, resulting in brighter clouds and stronger ERFaci$_{SW}$.
The relationship between aerosols, LWP and cloud fraction (Albrecht, 1989), however, is highly uncertain, varies
regionally (Sato et al., 2018), and is influenced by processes that are buffered over multiple spatiotemporal scales
(Stevens and Feingold, 2009). Additionally, E3SMv2's CFODD slope ("CNTL" simulation) agrees with MODIS-
CloudSat within uncertainty, indicating that droplet collection efficiency is well-represented according to CFODD
analysis.




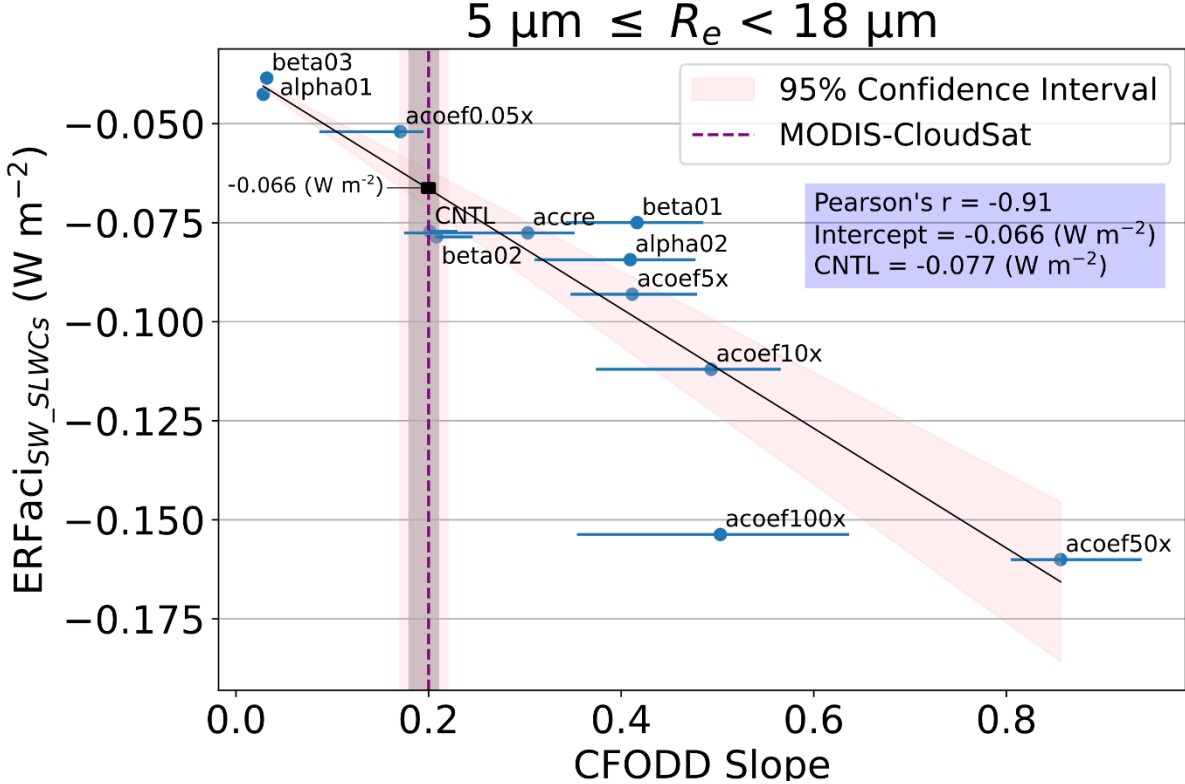

**Figure 3.** Linear regression between E3SMv2 ERFaci$_{SW\_SLWCs}$ and CFODD slopes, generated from SLWCs with MODIS $R_e$ between 5 and 18 µm, in 12 PD autoconversion and accretion sensitivity experiments. ERFaci$_{SW\_SLWCs}$ values reflect the SLWCs represented in the corresponding CFODD (i.e., with $R_e$ corresponding to the CFODD $R_e$ bin). Results show a strong negative correlation between E3SMv2 ERFaci$_{SW\_SLWCs}$ and CFODD slopes. We constrain the ERFaci$_{SW}$ by predicting the ERFaci$_{SW\_SLWCs}$ value at the reference MODIS-CloudSat $5 \leq R_e < 18$ µm CFODD slope (purple dashed line) from the linear regression (intercept shown in blue box). The constrained ERFaci$_{SW}$ value is decreased by $14 \pm 6\%$ in magnitude compared to the CNTL simulation. Error bars represent 1-sigma error estimated from RANSAC-fit bootstrapping (Sect. 2). Grey and pink shaded regions indicate the 68 and 95% confidence intervals for the MODIS-CloudSat CFODD slope, respectively. Labels indicate the sensitivity experiment names (Table 1).

In Figure 3, we constrain ERFaci$_{SW}$ due to autoconversion uncertainties using the linear regression between the simulated CFODD slopes and ERFaci$_{SW\_SLWCs}$. ERFaci$_{SW}$ and ERFaci$_{SW\_SLWCs}$ values were calculated following Ghan et al. (2013), which considers the difference in TOA radiative flux between the PD and PI experiments, neglecting direct forcing of aerosols (see Sect. 2.4 for details). We estimated the constrained value of ERFaci$_{SW\_SLWCs}$ at the intercept of the linear relationship with the observed MODIS-CloudSat CFODD slope (Fig. 4).The ERFaci$_{SW\_SLWCs}$

predicted by the linear regression at the MODIS-CloudSat slope value is -0.066 W m$^{-2}$, a 14 ± 6% decrease in
magnitude compared to the ERFaci$_{SW\_SLWCs}$ value predicted by the E3SMv2 CNTL simulation (-0.077 W m$^{-2}$).
E3SMv2's total ERFaci (-1.50 Wm$^{-2}$), inclusive of all cloud types and the longwave forcing component, falls within
the IPCC AR6 'very likely' range for ERFaci (-1.0 ± 0.7 Wm$^{-2}$). The shortwave component of ERFaci is significantly
larger than longwave in CMIP6 models (e.g., multimodel means of -0.91 and +0.10 W m$^{-2}$, respectively, as reported
in Smith et al. 2020). Thus, our results indicate that eliminated the bias in ERFaci$_{SW}$ due to autoconversion
uncertainties would decrease the magnitude of ERFaci$_{SW}$ and bring the predicted total ERFaci closer to the median
IPCC ERFaci value (Forster et al., 2021).

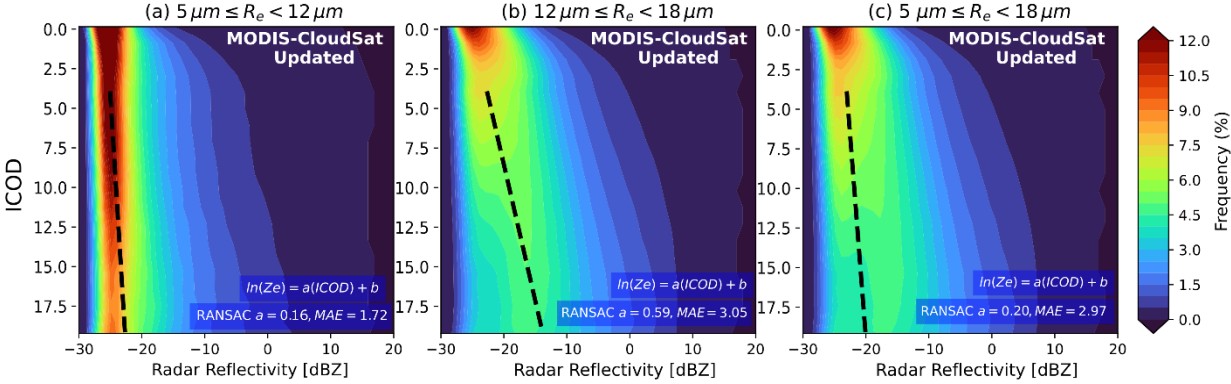

**Figure 4.** CFODDs for subset of SLWCs with max CloudSat reflectivity < 20 dBZ and COT < 20, June 2006 – April 2011, binned
by MODIS R$_e$ from updated reference MODIS-CloudSat observations analysis (a-b), and with combined "small" and "medium"
R$_e$ SLWCs in (c). RANSAC linear regressions were applied to the CFODD at 4 ≤ ICOD ≤ 20 to estimate droplet collection
efficiencies. RANSAC slopes and Median Absolute Error (MAE) values are shown in blue boxes.
As ERFaci$_{SW}$ is the result of many cloud processes, the updated CFODD analysis should be interpreted as a constraint
on the component of ERFaci$_{SW}$ that is modulated by droplet collection efficiency due to autoconversion. In other
words, the updated CFODD analysis shows the change in ERFaci$_{SW}$ one would expect if the bias in ERFaci$_{SW}$ due to
a specific process representation affecting droplet collection efficiency were eliminated. Base cloud processes that are
independent of aerosol also contribute significantly to ERFaci estimates (Mülmenstädt et al., 2020). Autoconversion
perturbations affect base cloud state (e.g., LWP, cloud fraction) and could, for example, cause stronger ERFaci by
increasing cloud amount rather than increasing the impact of ACI on SW radiative forcing. Jing et al. (2019) evaluated
different autoconversion parameterization schemes in an ESM using the CFODD analysis described in Michibata et

al. (2019b) and found that the autoconversion scheme that yielded the best warm rain representation predicted a significantly stronger ERFaci that exceeded the uncertainty range of the IPCC AR5 and canceled out much of the warming trend of the last century. The conflict between process representation and ERFaci predictions in Jing et al. (2019) underscore a challenge with process-based constraints: improving the representation of a process can result in adverse outcomes to climate prediction due to compensating biases in the model. This challenge is particularly troublesome for constraints on processes like autoconversion that affect the base cloud state because decreasing autoconversion rates can increase total cloud amount, which can yield stronger ERFaci. Thus, a decreased autoconversion rate may improve precipitation outcomes in an ESM that presents the common "too frequent" warm rain bias (e.g., Stephens et al., 2010), yet cause improbably strong ERFaci. Our results show, however, that decreased autoconversion rates result in weaker $ERFaci_{SW\_SLWCs}$ (Fig. 3), demonstrating that the base cloud state issue presented in prior studies of autoconversion is not a dominant factor contributing to the $ERFaci_{SW}$ of warm rain processes in E3SMv2.

Figure 5a shows the linear relationship between $ERFaci_{SW\_SLWCs}$ normalized by the PI SW Cloud Radiative Effect (SWCRE), which represents the fraction of ERFaci that is independent of base cloud state changes, and CFODD slope. The correlation coefficient in Fig. 5a (Pearson's R = 0.74) is decreased compared to Fig. 3 (Pearson's R = -0.91). However, comparing the negative correlations between CFODD slope and PI SLWC cloud fraction (Fig. 5b; Pearson's R = -0.64) and LWP (Fig. 5c; Pearson's R = -0.89) with Fig. 3, the $ERFaci_{SW\_SLWCs}$ increases in magnitude as LWP and cloud fraction decrease, further demonstrating that the contribution of base cloud state to $ERFaci_{SW\_SLWCs}$ is relatively minor. The decreased correlation coefficient in Fig. 5a could also be influenced by poor sampling statistics in the "acoef100x" experiment. The acoef100x was the only one of six experiments involving perturbations of the "A" coefficient in KK2000 (Table 1; Sect. 2.4) in which the CFODD slope did not increase with an increase in magnitude of the "A" coefficient. Given the significant decrease in SLWC cloud fraction in this experiment compared to the others (Fig. 5b, Table S2), the CFODD slope result may be affected by insufficient sample size, an additional uncertainty of the CFODD linear regression that is not reflected in the bootstrapping-based uncertainty estimate (Sect. 2).

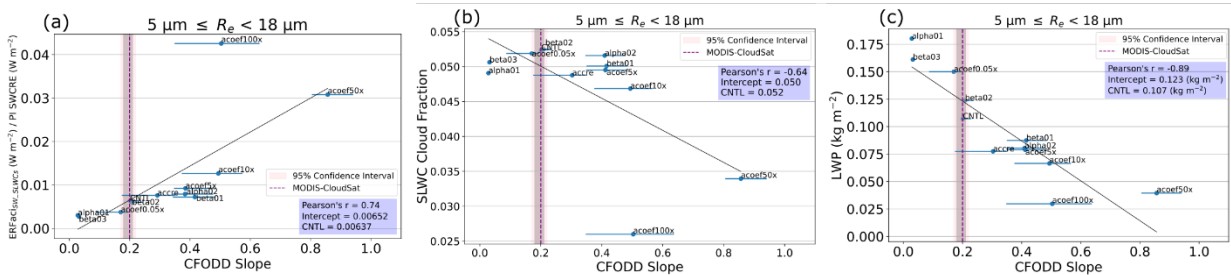

446

**Figure 5.** Linear regression between (a) E3SMv2 $ERFaci_{SW\_SLWCs}$ normalized by SWCRE, (b) SLWC cloud fraction, (c) SLWC LWP and CFODD slopes in 12 PD autoconversion and accretion sensitivity experiments, calculated for SLWCs with MODIS $R_e$ between 5 and 18 µm. $ERFaci_{SW\_SLWCs}$ values reflect the SLWCs represented in the corresponding CFODD (i.e., with $R_e$ corresponding to $5 < R_e < 18$ µm). Error bars represent 1-sigma error estimated from RANSAC-fit bootstrapping (Sect. 2). Grey and pink shaded regions indicate the 68 and 95% confidence intervals for the MODIS-CloudSat CFODD slope, respectively. Labels indicate the sensitivity experiment names (Table 1).

While we derive a constraint for $ERFaci_{SW}$ using the combined small and medium $R_e$ CFODDs, when the $R_e$ subsets are considered individually, they show distinct contributions to $ERFaci_{SW\_SLWCs}$. Fig. S7 shows that SLWCs with small $R_e$ have a negative $ERFaci_{SW\_SLWCs}$, but that SLWCs in the medium and large $R_e$ subsets have positive $ERFaci_{SW\_SLWCs}$ values. This indicates that the dominant effect of aerosols on shortwave radiative forcing in the medium and large SLWC subsets is decreased cloud fraction, which is reflected in the decreased SLWC sample sizes in the PD simulations compared to PI (Fig. S8, S9). The negative linear relationship between $ERFaci_{SW\_SLWCs}$ and CFODD slope in the medium and large $R_e$ subsets indicates that increasing droplet collection efficiency partially counteracts the decrease in cloud fraction due to aerosol. The small $R_e$ SLWCs, however, show a positive correlation between $ERFaci_{SW}$ and CFODD slope, indicating that $ERFaci_{SW}$ weakens as autconversion rates increase, likely due to decreased precipitation suppression susceptibility in this subset. The combined small and medium CFODD and $ERFaci_{SW\_SLWCs}$, therefore, represent the convolution of two populations with differing $ERFaci_{SW}$ sensitivities to autoconversion perturbations. We chose to constrain $ERFaci_{SW}$ using the combined small and medium CFODD and $ERFaci_{SW\_SLWCs}$ due the correlation performance and the dearth of large $R_e$ SLWCs in E3SMv2. However, constraints for $ERFaci_{SW}$ could potentially be derived for each individual $R_e$ subset or various combinations thereof, depending on the distribution of SLWCs among the $R_e$ size bins and their contribution to the host model's ERFaci. Considering that constrained $ERFaci_{SW}$ increases in magnitude with increasing $R_e$ in Fig. S7 the underrepresentation of SLWCs

with large $R_e$ in E3SMv2 represents a compensating bias, without which the total ERFaci bias would be even larger
compared to IPCC AR6.

**4.2 Limitations of CFODD-based constraint on ERFaci**

There are multiple limitations to the CFODD analysis that should be considered in its application as a constraint for
ERFaci. First, droplet collection is not explicitly represented in ESMs with bulk microphysical schemes such as
E3SMv2, but is instead implicit in an amalgamation of process and drop size distribution parameterizations controlling
the evolution of the cloud and precipitation. Delving into the impact of these individual processes on CFODD-based
constraint of ERFaci is a good target of future work, while autoconversion modulation of ERFaci was the primary
focus here. Furthermore, simulated radar reflectivity is highly sensitive to particle size distribution assumptions in the
forward simulator (e.g., Bodas-Salcedo et al., 2011; J. Wang et al., 2021). The host model, therefore, could represent
warm rain microphysical processes with high fidelity but still produce biased CFODD profiles when compared with
observations.  In COSPv2.0, the CloudSat simulator calculates size distributions from an assumed distribution (e.g.,
log-normal, gamma, exponential) as well as mass-mixing ratios, precipitation fluxes, and gridbox-mean $R_e$ from the
host model. Default COSPv2.0 size distributions were used in this study: log-normal for large-scale stratiform and
convective cloud liquid, and exponential for large-scale stratiform and convective cloud rain. The CFODD analysis
itself is subject to multiple uncertainties, including the use of simple adiabatic and condensational growth assumptions
to scale MODIS COT to ICOD. These assumptions result in a vertical distribution of optical depth, mass
concentrations and particle size that may not reflect reality. For example, in the CFODD, particle size and mass
concentration are assumed to monotonically increase with height, yet in the real cloud, particle sizes may decrease
near the cloud top due to evaporation and entrainment (Suzuki et al., 2010). The uncertainties from assumed
hydrometeor size distributions and CFODD construction should be carefully considered when using the CFODD to
evaluate model droplet collection efficiencies against observations and in the application as an ERFaci constraint.
Simulated reflectivity biases affect the evaluation of the model CFODD slope against the observational CFODD slope
and thus affect the estimation of ERFaci bias.
A few additional limitations on CFODD analysis are imposed by biases in E3SMv2 SLWC representation. The ERFaci
constraint is restricted to the small and medium $R_e$ CFODDs because of the underrepresentation of SLWCs with large
$R_e$. SLWCs with medium $R_e$ are also underrepresented in E3SMv2, further limiting the CFODD analysis of E3SMv2
ERFaci because process perturbations are limited to the extent that they do not significantly reduce the number of
SLWCs with medium $R_e$. The high reflectivity near cloud top at ICOD < 4 in E3SMv2 CFODDs presents another
limitation. SLWCs with COT < 4 represent a significant fraction of the SLWC population in both A-Train and
E3SMv2 (Fig. 2), so including optically thin SLWCs in the linear regression would likely affect the CFODD slope
and droplet collection efficiency estimates.
Despite these limitations and the uncertainty associated with estimates of droplet collection efficiency from simulated
radar reflectivity, CFODD analysis offers a highly desired process-oriented constraint on ERFaci due to warm rain
processes. In E3SMv2, the CFODD slope exhibits the expected behavior in response to autoconversion perturbations:
slope increases with perturbations that increase the autoconversion rate and decreases with perturbations that decrease
the autoconversion rate. Our results also show that the model ERFaci$_{SW}$ is highly sensitive to the processes that the
CFODD represents, enabling the constraint of ERFaci$_{SW}$ against the CFODD slope derived from MODIS-CloudSat
cloud retrievals. Prior studies have demonstrated that radar reflectivity biases can be partially mitigated by bringing
the forward simulator into better agreement with the host model's microphysics parameterization and subgrid
variability (Song et al., 2018b; J. Wang et al., 2021). Modified versions of COSP featuring improved consistency with
E3SM are to be implemented in future E3SM model versions, which will decrease the uncertainties associated with
CFODD analysis of E3SM.

## 5 Summary

In this study, we present an updated CFODD analysis and demonstrate how it can be applied to ESMs as a process-
oriented constraint on ERFaci. When E3SMv2's CFODD slope is constrained by MODIS-CloudSat retrievals,
E3SMv2's ERFaci$_{SW}$ is reduced by $14 \pm 6\%$. Demonstrated here as a constraint on the component of ERFaci$_{SW}$ that is
modulated by autoconversion, CFODD analysis represents a highly desirable constraint on a process, circumventing
the equifinality issue that bedevils atmospheric state variable-based approaches (Mülmenstädt et al., 2020).
Limitations of CFODD-based constraint of ERFaci include the implicit representation of droplet collection efficiency
in many ESMs, including E3SMv2, the sensitivity of simulated radar reflectivity to droplet size distribution
representations and simplifying assumptions applied to construct the CFODD (e.g., adiabatic-condensational growth).
While this study focuses on autoconversion, future studies should apply CFODD analysis to other microphysical
processes that affect droplet collection efficiency (e.g., accretion, droplet breakup, evaporation) to generate additional
ERFaci constraints.
Several updates to the WRDs package in COSPv2.0 were made to support the application of CFODD analysis to
ESMs. In addition to the original WRDs diagnostics featuring relative frequencies of SLWCs by precipitation intensity
and the CFODD by $R_e$, we have implemented additional diagnostics in the WRDs that include all-sky SLWC
frequency maps and MODIS SLWC COT distributions for CFODD sampling statistics. Other updates include the
estimation of CFODD slopes using Random Sample Consensus robust linear regression and changes to the SLWC
detection schemes for better comparison between observations and satellite simulators.
In addition to the modifications of the WRDs described above, the MODIS and CloudSat observational reference data
has been updated for consistency with COSPv2.0 SLWC detection. SLWC detection is increased 5-fold in the updated
reference data. The increase in SLWC sampling also significantly affected the CFODD distributions and consequently,
the A-Train reference droplet collection efficiency at large $R_e$ ($18\,\mu m \le R_e < 30\,\mu m$). The updated WRDs showed that
droplet collection efficiencies in E3SMv2 are decreased compared to observations and SLWCs with small MODIS $R_e$
($5\,\mu m \ge R_e > 12\,\mu m$) are overrepresented. The E3SMv2 CFODD results also show reflectivities exceeding 0 dBZ near
cloud top at $2 < ICOD < 4$ yet relatively low reflectivities at $ICOD > 5$. The unreasonably high reflectivities near cloud
top may indicate artifacts due to inconsistencies between E3SMv2 outputs and COSPv2.0 inputs to the CloudSat
simulator. This issue motivates further investigation in future studies involving applications of the CloudSat simulator
to E3SM. The updates described herein have increased the WRDs' utility for evaluating model warm rain process
representation and support the analysis needed to derive a constraint on ERFaci from CFODD analysis. Through an
evaluation of E3SMv2, we demonstrate that the updated WRDs illuminate specific biases in SLWC representation
and provide contextual sampling statistics that are critical for interpreting CFODD results and thus, for future
applications of this observational constraint on ERFaci.

*Code and Data Availability:* The CloudSat and MODIS data products are available from the CloudSat Data Processing
Center at CIRA/Colorado State University (https://www.cloudsat.cira.colostate.edu/; last access: June 28, 2023). The
reference A-Train data used in this study is available here: https://doi.org/10.5281/zenodo.8384180. The modified
source code of COSPv2.0 is available here: https://doi.org/10.5281/zenodo.8371120 and the E3SMv2 source code is
available here: https://github.com/E3SM-Project/E3SM (last access: September 27, 2023). The python package for
the two-dimensional Kolmogorov-Smirnov test applied in this study is available here
(https://github.com/syrte/ndtest/tree/master; last access: June 28, 2023). The python package scikit-learn was used for
robust linear regression analysis (https://scikit-learn.org/stable/; last access: June 28, 2023).
*Author contributions:* CMB led the project, developed the additional WRDs diagnostics in this study, performed the
model simulations and wrote the manuscript. PLM provided critical project guidance and support for modeling and
analysis. MWC led the A-Train observations analysis and provided guidance on additional WRDs diagnostics
development. AV provided input on CFODD analysis applications. JM provided guidance on ERFaci analysis. TM
and KS provided guidance on WRDs applications. All authors contributed to writing the manuscript.
*Competing Interests:* At least one of the (co-)authors is a member of the editorial board of Atmospheric Chemistry
and Physics.
*Acknowledgements*: The study was supported as part of the Enabling Aerosol–cloud interactions at GLobal
convection-permitting scalES (EAGLES) project (project no. 74358) sponsored by the United States Department of
Energy (DOE), Office of Science, Office of Biological and Environmental Research (BER), Earth System Model
Development (ESMD) program area. The Pacific Northwest National Laboratory (PNNL) is operated for the DOE by
the Battelle Memorial Institute under Contract DE-AC05-76RL01830. The research used high-performance
computing resources from the PNNL Research Computing, the BER Earth System Modeling program's Compy
computing cluster located at PNNL, and resources of the National Energy Research Scientific Computing Center
(NERSC), a U.S. Department of Energy Office of Science User Facility located at Lawrence Berkeley National
Laboratory, operated under Contract No. DE-AC02-05CH11231, using NERSC awards ALCC-ERCAP0025938 and
BER-ERCAP0024471.
*Financial support*.  This study was funded by the U.S. Department of Energy, Office of Science, Office of Biological
and Environmental research, Earth System Model Development (ESMD) program area (project nos. 74358). KS and
TM were supported by the Japan Society for the Promotion of Science KAKENHI (Grant JP19H05669), MEXT
program for the Advanced Studies of Climate Change Projection (SENTAN) (Grant JPMXD0722680395), and the
Environment Research and Technology Development Fund (S-20) (Grant JPMEERF21S12004) of the Environmental
Restoration and Conservation Agency. TM was supported by the JST FOREST Program (Grant JPMJFR206Y),
and the Japan Society for the Promotion of Science KAKENHI (Grant JP 23K13171).

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
