# Peer review of "Droplet collection efficiencies estimated from satellite retrievals"

_EGUsphere, 2023_

## Author Response (AR1)

Response to 'Comment on egusphere-2023-2161', updated with line-by-line changes. Line numbers reflect those of the track-changes version of the revised manuscript.

RC1: "However, at the moment, there is a fundamental methodological issue that I cannot understand and therefore I cannot provide a full review until the methodology is clarified. Specifically, note Figure 3. There is something basic that I'm missing. Why are the ERF_ACI_SW values so small? I would expect negative values near 1 Wm^-2. Also why does ERF_ACI_SW differ between figures 3, and S7. Some of the ERF_ACI_SW are even positive in S7! I would think that you have 13 experiments so you should have 13 fixed values of ERF_ACI_SW. It is almost like the total ERF_ACI has been decomposed into components in some way, however I can find no description of this in the text. Until the methodology is further explained I can not complete this review. In addition I have a few specific comments below."

Reposting our AC1 response here for easy access, updated with line-by-line changes.

We thank the reviewer for bringing attention to the clarification needed for the ERFaci values in Fig. 3 and Fig. S7 and for the helpful specific comments. The ERFaci$_{SW}$ values in Fig. 3 are smaller in magnitude than an expected value for total ERFaci because they reflect the ERFaci$_{SW}$ calculated for only the cloud samples featured in the CFODDs (i.e., the SLWCs with $5\,\mu m \le R_e < 18\,\mu m$). These SLWC samples represent ~8 % of the total simulated cloud fraction according to the MODIS simulator. The reason for the different ERFaci values between Fig. 3 and the subplots in Fig. S7 is that ERFaci values in each plot reflects the subset of SLWC samples in the MODIS $R_e$ bin indicated by the plot title. If you sum the ERFaci$_{SW}$ values for each simulation in Fig. S7 (a) ($5\,\mu m \le R_e < 12\,\mu m$) and (b) ($12\,\mu m \le R_e < 18\,\mu m$), the resulting values are equivalent to the values shown in Fig. 3 ($5\,\mu m \le R_e < 18\,\mu m$). The positive ERFaci$_{SW}$ values in the medium and large $R_e$ bins partially offset the negative ERFaci$_{SW}$ values in the smallest $R_e$ bin, resulting in an ERFaci$_{SW}$ that is smaller in magnitude than for the $5\,\mu m \le R_e < 12\,\mu m$ bin alone. This may be a result of compensating biases in E3SMv2 where the positive ERFaci$_{SW}$ predicted for medium and large $R_e$ bins is offset by simulating too many SLWCs in the small $R_e$ bin. The point that calculation of ERFaci is made for SLWCs within the specified MODIS $R_e$ range is mentioned on L288, but we now specify that ERFaci is calculated for SLWCs in the y-axis labels (e.g., ERFaci$_{SW\_SLWCs}$), the figure captions and throughout the text at the locations described below.

Abstract L24: We estimate the shortwave component of ERFaci (ERFaci$_{SW}$), constrained by MODIS-CloudSat, by calculating the intercept of the linear association between the E3SMv2 ERFaci$_{SW}$ of SLWCs and the CFODD slopes, using the MODIS-CloudSat CFODD slope as a reference.

L366: To demonstrate the potential of the CFODD analysis described above for constraining ERFaci,$_{SW}$ due to warm rain processes, we performed 12 experiments featuring variations of E3SMv2's autoconversion and accretion parameterizations, computing ERFaci$_{SW}$ for the SLWC samples represented in each CFODD and the corresponding $R_e$ bin (hereafter, "ERFaci$_{SW\_SLWCs}$") following Ghan (2013; see Sect. 2.4).

L375: Figure 3 shows a strong negative correlation between E3SMv2 ERFaci$_{SW\_SLWCs}$ with "small" or "medium" $R_e$ (i.e., $5 \le R_e < 18\,\mu m$) and the corresponding combined $5 \le R_e < 18\,\mu m$ CFODD slope (Pearson's R = -0.91).

L448: The correlation between ERFaci$_{SW\_SLWCs}$ and CFODD slope is stronger in the combined CFODDs relative to the CFODDs considered separately (Fig. S7, also see discussion below).

Supplement Fig. S7 caption: **Figure S7.** Linear regression between E3SMv2 ERFaci$_{SW\_SLWCs}$ and CFODD slopes in 12 PD autoconversion and accretion sensitivity experiments, binned by MODIS $R_e$. ERFaci$_{SW\_SLWCs}$ values reflect the SLWCs represented in the corresponding CFODD (i.e., with $R_e$ corresponding to the CFODD $R_e$ bin).

Figure S7 y-axis labels were also updated from ERFaci$_{SW}$ to ERFaci$_{SW\_SLWCs}$.

Figure 3 caption: Linear regression between E3SMv2 ERFaci$_{SW\_SLWCs}$ and CFODD slopes, generated from SLWCs with MODIS $R_e$ between 5 and 18 µm, in 12 PD autoconversion and accretion sensitivity experiments. ERFaci$_{SW\_SLWCs}$ values reflect the SLWCs represented in the corresponding CFODD (i.e., with $R_e$ corresponding to the CFODD $R_e$ bin). Results show a strong negative correlation between E3SMv2 ERFaci$_{SW\_SLWCs}$ and CFODD slopes. We constrain

the $ERFaci_{SW}$ by predicting the $ERFaci_{SW\_SLWCs}$ value at the reference MODIS-CloudSat $5 \leq R_e < 18$ µm CFODD slope (purple dashed line) from the linear regression (intercept shown in blue box).

Figure 3 y-axis labels also updated from $ERFaci_{SW}$ to $ERFaci_{SW\_SLWCs}$.

Fig. S9 caption: Linear regression between E3SMv2 $ERFaci_{SW\_SLWCs}$ normalized by SWCRE and CFODD slopes in 12 PD autoconversion and accretion sensitivity experiments, generated from SLWCs with MODIS $R_e$ between 5 and 18 µm. $ERFaci_{SW\_SLWCs}$ values reflect the SLWCs represented in the corresponding CFODD (i.e., with $R_e$ corresponding to $5 < R_e < 18$ µm).

Fig. S9 (now Fig. 5) y-axis labels were updated from $ERFaci_{SW}$ to $ERFaci_{SW\_SLWCs}$.

L400: Results show a strong negative correlation between E3SMv2 $ERFaci_{SW\_SLWCs}$ and CFODD slopes.

L408: In Figure 3, we constrain $ERFaci_{SW}$ due to autoconversion uncertainties using the linear regression between the simulated CFODD slopes and $ERFaci_{SW\_SLWCs}$. We estimated the constrained value of $ERFaci_{SW\_SLWCs}$ at the intercept of the linear relationship, using the observed MODIS-CloudSat CFODD slope (Fig. S8) as a reference.

L442: Our results show, however, that decreased autoconversion rates result in weaker $ERFaci_{SW\_SLWCs}$ (Fig. 3) [...]

L446: Figure 5a shows the linear relationship between $ERFaci_{SW\_SLWCs}$ normalized by the PI SW Cloud Radiative Effect (SWCRE), [...]

L449: However, comparing the negative correlations between CFODD slope and PI cloud fraction (Fig. S10; Pearson's R = -0.64) and LWP (Fig. S11; Pearson's R = -0.89) with Fig. 3, the $ERFaci_{SW\_SLWCs}$ increases in magnitude as LWP and cloud fraction decrease, further demonstrating that the contribution of base cloud state to $ERFaci_{SW\_SLWCs}$ is relatively minor.

L466: While we derive a constraint for $ERFaci_{SW}$ using the combined small and medium $R_e$ CFODDs, when the $R_e$ subsets are considered individually, they show distinct contributions to $ERFaci_{SW\_SLWCs}$.

L467: Fig. S7 shows that SLWCs with small $R_e$ have a negative $ERFaci_{SW\_SLWCs}$, but that SLWCs in the medium and large $R_e$ subsets have positive $ERFaci_{SW\_SLWCs}$ values.

L471: The negative linear relationship between $ERFaci_{SW\_SLWCs}$ and CFODD slope in the medium and large $R_e$ subsets indicates that increasing droplet collection efficiency partially counteracts the decrease in cloud fraction due to aerosol.

L473: The small $R_e$ SLWCs, however, show a positive correlation between $ERFaci_{SW}$ and CFODD slope, indicating that $ERFaci_{SW}$ weakens as autoconversion rates increase, likely due to decreased precipitation suppression susceptibility in this subset.

L477: The combined small and medium CFODD and $ERFaci_{SW\_SLWCs}$, therefore, represent the convolution of two populations with differing $ERFaci_{SW}$ sensitivities to autoconversion perturbations.

L479: We chose to constrain $ERFaci_{SW}$ using the combined small and medium CFODD and $ERFaci_{SW\_SLWCs}$ due the correlation performance and the dearth of large $R_e$ SLWCs in E3SMv2.

Responses to the reviewer's specific comments are below:

"Abstract: Abstract should be less than 250 words according to the ACP guidelines"

The abstract word count was reduced to 248 words.

"The following paper is very similar in scope and methods and should be cited in the introduction: Takahashi, H., A. Bodas-Salcedo, and G. Stephens, 2021: Warm Cloud Evolution, Precipitation, and Their Weak Linkage in HadGEM3: New Process-Level Diagnostics Using A-Train Observations. J. Atmos. Sci., 78, 2075–2087, https://doi.org/10.1175/JAS-D-20-0321.1."

Agreed that this reference is highly relevant to our manuscript. This reference was added to the introduction:

L103: Takahashi et al. (2021) proposed an updated CFODD analysis in which $R_e$ thresholds are defined by quartile distributions of SLWC samples rather than the traditional CFODD $R_e$ thresholds to focus evaluation on warm rain process representation rather than the bias in $R_e$ distribution.

"Line 125: is there a reference for RANSAC? I think this is a python package but will not be familiar to the average reader."

Good point. The following reference for RANSAC was added:

L142: Fischler, M. A. and R. C. Bolles (1981). "Random sample consensus: a paradigm for model fitting with applications to image analysis and automated cartography." Commun. ACM **24**(6): 381–395.

"Line 221: MODIS provides 3 different effective radius values from the 1.6, 2.1, and 3.7 micron channels. 2.1 micron is the 'standard' product. Which one are you using?"

Yes, we are using the standard 'Cloud_Effective Radius' and 'Cloud_Effective Radius_PCL' products retrieved from the 2.1 micron channel, consistent with the MODIS simulator used in the WRDs. This detail has been added to the text.

L251: Standard MODIS products from the 2.1 µm channel were used for $R_e$, consistent with the simulated MODIS $R_e$ used in the WRDs.

"Line 260 and Figure 1: Is a SLWC defined as a cloud with optical depth > 0.3 and radar reflectivity > -30 dBZ? If this is correct can you mention this in the text. In this case I would assume that the identification of detectable cloud is nearly entirely limited by the radar reflectivity threshold whereas the optical depth threshold probably was the dominant limiting factor in the earlier study."

Yes, this is correct. The following was added to L119: "The criteria for SLWC detection in A-Train observations and E3SMv2-COSP are provided in the Supplement Table 1 and include CloudSat reflectivity ≥ -30 dBZ, MODIS liquid COT > 0.3, and cloud top temperature ≥ 273 K."

"Figure 3: You should include an uncertainty range around the linear fit as gray shading. You should quote that uncertainty in the text."

Agreed. An uncertainty estimate for the linear fit was added to Fig. 3. The following text on the uncertainty analysis was added to Sect. 2.4:

L290: A constraint on ERFaci$_{SW}$ was calculated from the linear regression between E3SMv2 CFODD slopes and ERFaci$_{SW}$, using the MODIS-CloudSat CFODD slope as a reference. A 95% confidence interval for the linear fit was estimated by bootstrapping the linear regression within the uncertainty of the CFODD slopes. CFODD slope values were randomly sampled 1000 times within their 1-sigma error and repeatedly regressed with ERFaci$_{SW}$. The original data (i.e., RANSAC CFODD slope values and corresponding ERFaci$_{SW}$ values) were additionally resampled with replacement to generate a distribution of coefficients for the ordinary least squares (OLS) regression. The 95% confidence interval for the linear fit was then calculated from the combined linear regression coefficient distributions to reflect uncertainty from both the OLS fit and the CFODD slopes.

"Line 350: This idea of adjusting the estimate of the ERF_ACI based on this linear fit seems bizarre to me. The CNTRL experiment already agrees with the observed CFODD slope almost exactly. So to me the story of this paper is that this version of the model agrees very well with this particular metric which suggests that the auto conversion process is been simulated with relatively good fidelity."

Agreed that the CNTL experiment CFODD slope agrees with the observed CFODD slope within uncertainty. This study presents a constraint on the component of ERFaci$_{SW}$ that is affected by droplet collection efficiency due to autoconversion, but in each simulation, the predicted ERFaci$_{SW\_SLWCs}$ is affected by many other processes, so we don't expect that the ERFaci$_{SW\_SLWCs}$ should agree with the constrained value whenever the droplet collection efficiency

estimate agrees. For example, cloud top entrainment also affects cloud liquid water path and consequently, ERFaci (Ackerman et al., 2004). The results show that the component of ERFaci$_{SW}$ that is affected by droplet collection efficiency due to autoconversion is overestimated by 14% ± 6% compared to the constrained value.

The relationship between this process-based constraint on ERFaci$_{SW}$, overall ERFaci and other cloud processes is described in a few places, including the paragraph starting on L400, L409 and L496.

Response to 'Reply on AC1':

"I'm a little uncomfortable with the title phrasing of 'droplet collection efficiency'. The paper looks at a proxy for efficiency but doesn't quantify the collection efficiency in a direct way. Consider a more accurate title."

We have updated the title to "Droplet collection efficiencies inferred from satellite retrievals constrain effective radiative forcing of aerosol-cloud interactions". The title term "inferred" is consistent with Suzuki et al. (2010).

Abstract line 26: '*When E3SMv2's droplet 27 collection efficiency is constrained to agree with the A-Train retrievals, ERFaciSW is reduced by 13% in magnitude*' Consistent with my previous statement I don't think this is the correct takeaway. I would state the model droplet collection efficiency is consistent with the observations lending credence to the estimate of the ERFaciSW in the model.

After adding the uncertainty analysis to the ERFaci$_{SW\_SLWCs}$ vs. CFODD slope fit, we find that the E3SMv2 ERFaci$_{SW}$ estimate does not agree with the constrained value within uncertainty and should be reduced 14 ± 6% in magnitude. See response to prior comment on the implications of CFODD slope agreement on the ERFaci$_{SW}$ constraint.

While I appreciate the sentiment of using a process-oriented constraint instead of state variables to evaluate the representation of ERFaci I have a lot of concern of relying on a single metric to make any sweeping conclusions about the fidelity of the simulations. For example, while the current version of this model agrees with this particular metric it is telling (that like many models) it does not reproduce the historical temperature trend (Golaz et al., 2022). To address my concerns, I think the paper would benefit from some analysis of some state variables. At the very least it would be nice to see an analysis of the TOA reflected shortwave biases relative to EBAF for the 13 simulations. Furthermore, since you've run the MODIS simulator it would be easy to evaluate the CF and LWP as well.

We agree that the CFODD-based constraint alone cannot be used to assess model fidelity. However, the point of this paper is to demonstrate a constraint on just the component of ERFaci$_{SW}$ related to droplet collection efficiency due to a single warm rain process representation rather than to constrain ERFaci overall. For this reason, we qualify the interpretation of our result that the constrained value should be reduced 14 ± 6% with phrasing to indicate that this is indicative only of what would happen if one were to correct bias in ERFaci$_{SW}$ due to autoconversion, according to this CFODD-based metric (e.g., L27 of abstract). If we were to perform the same analysis with another process affecting droplet collection efficiency (e.g., accretion), we would expect a different result, which would demonstrate the impact of that process on ERFaci$_{SW}$ bias. In other words, the CFODD analysis-based constraint is a tool that shows what change in ERFaci you would expect if the bias in ERFaci due to that specific process were corrected. A process-oriented constraint only constrains the component of ERFaci related to the process, not overall ERFaci.

To make this point clear, we have added the following text:

L63: Process-oriented constraints on ERFaci are useful for quantifying the sensitivity of ERFaci to a specific process or constraining the component of ERFaci that is affected by a process, rather than for constraining ERFaci overall (Mülmenstädt and Feingold, 2018).

L109: The updated CFODD analysis is demonstrated here as a constraint on the component of ERFaci$_{SW}$ that is affected by droplet collection efficiency due to autoconversion.

L530: Demonstrated here as a constraint on the component of ERFaci$_{SW}$ that is modulated by autoconversion, CFODD analysis represents a highly desirable constraint on a process, circumventing the equifinality issue that bedevils atmospheric state variable-based approaches (Mülmenstädt et al., 2020).

To the reviewer's follow-on point that E3SMv2 does not reproduce the historical temperature trend, ERFaci agreement with any constraint is not an indication that the surface temperature response to climate forcing should agree with historical trends because there are many other components of the climate system that modulate the surface temperature. Mulcahy et al. (2023) demonstrated that UKESM1 and UKESM1.1 produce very different historical surface temperature evolution despite modest changes in aerosol ERF and conclude that aerosol ERF is unlikely to explain the change in surface temperature evolution. Furthermore, CE3SM2 has a similar aerosol ERF to E3SvMv1 (-1.67 W m$^{-2}$ and -1.65 W m$^{-2}$, respectively) (Golaz et al., 2019; Gettelman et al., 2019), but produces a very different historical surface temperature trend and does not show the rapid cooling in 1960-1970 seen in E3SMv1 (Danabasoglu et al., 2020).

We think that an evaluation of e.g., LWP and CF and TOA radiation balances against EBAF in our 13 experiments would detract from the main point of this paper -- to demonstrate a process-oriented constraint on ERFaci. These state variables are the result of many processes, whereas we are targeting a single warm rain process and constraining the component of ERFaci affected by the process representation. Zhang et al. (2024) presents an evaluation of E3SMv2 cloud state and cloud radiative effect using MODIS, ISCCP, MISR and CALIPSO, if these points are of interest.

"Finally, some discussion of these concerns, limitations of the study, and outlook for future research should also be added in the summary."

Agreed that a summary of limitations and future directions should be included in the summary section.

The following has been added to L533:

Limitations of CFODD-based constraint of ERFaci include the implicit representation of droplet collection efficiency in many ESMs, including E3SMv2, the sensitivity of simulated radar reflectivity to droplet size distribution representations and simplifying assumptions applied to construct the CFODD (e.g., adiabatic-condensational growth). While this study focuses on autoconversion, future studies should apply CFODD analysis to other microphysical processes that affect droplet collection efficiency (e.g., accretion, droplet breakup, evaporation) to generate additional ERFaci constraints.

Response to RC2:

We thank the reviewer for the helpful guidance on paper organization and suggestions for emphasis on the E3SMv2 control simulation performance.

"Main Comments: The manuscript provides an adequate description of the model and the satellite data that are used, including their limitations. I appreciate the authors' approach using CFODD to constrain the simulations, it is elegant and provide interesting results. Nevertheless, the CTRL simulation seems to agree well with the observations. This is a good result for the model and the control coefficients. Shouldn't this be the main conclusion of the study?"

We agree that the E3SMv2 CFODD slope agreed with MODIS-CloudSat within uncertainty, and have added the following to the text to emphasize this point:

L391: Additionally, E3SMv2's CFODD slope ("CNTL" simulation) agrees with MODIS-CloudSat within uncertainty, indicating that droplet collection efficiency is well-represented according to CFODD analysis.

However, we don't expect that the ERFaci$_{SW}$ should agree with the constrained value whenever the droplet collection efficiency estimate agrees. Our results demonstrate a constraint on the component of ERFaci$_{SW}$ that is affected by droplet collection efficiency due to autconversion, but the predicted ERFaci$_{SW}$ is affected by many other processes. If we were to apply this analysis to another process affecting droplet collection efficiency, e.g., accretion, we might see a different relationship between CFODD slope and ERFaci$_{SW}$. The constraint presented here tells us what change in ERFaci$_{SW}$ we would expect if we were to correct the bias in the process representation, according to this MODIS-CloudSat metric. We have added the following text to guide interpretation of the CFODD-based constraint on ERFaci:

L426: As ERFaci$_{SW}$ is the result of many cloud processes, the updated CFODD analysis should be interpreted as a constraint on the component of ERFaci$_{SW}$ that is modulated by droplet collection efficiency due to autoconversion. In other words, the updated CFODD analysis shows the change in ERFaci$_{SW}$ one would expect if the bias in ERFaci$_{SW}$ due to a specific process representation affecting droplet collection efficiency were corrected.

We have also added text to specify what the updated CFODD analysis constrains:

L63: Process-oriented constraints on ERFaci are useful for quantifying the sensitivity of ERFaci to a specific process or constraining the component of ERFaci that is affected by a process, rather than for constraining ERFaci overall (Mülmenstädt and Feingold, 2018).

L108: The updated CFODD analysis is demonstrated here as a constraint on the component of ERFaci$_{SW}$ that is affected by droplet collection efficiency due to autoconversion.

L530: Demonstrated here as a constraint on the component of ERFaci$_{SW}$ that is modulated by autoconversion, CFODD analysis represents a highly desirable constraint on a process, circumventing the equifinality issue that bedevils atmospheric state variable-based approaches (Mülmenstädt et al., 2020).

"Is there a physical meaning for choosing other coefficients, in this model or others? it would be interesting for discussion."

The coefficient values chosen for the experiments are bounded by those in past studies including KK2000, Wood (2005), and Kogan (2013). The main difference with the Kogan (2013) coefficient values is that they were derived from a large-eddy simulation (LES) with bin resolved microphysics for cumulus clouds, whereas the focus of Wood (2005) and KK2000 was stratocumulus. Kogan (2013)'s approach also used LES with bin-resolved microphysics whereas Wood (2005) derived autoconversion coefficients from observations. These details were added along with a missing Kogan (2013) reference.

L270: The Kogan (2013) coefficient values were derived from a large-eddy simulation (LES) with bin resolved microphysics for cumulus clouds, whereas the focus of Wood (2005) and KK2000 was stratocumulus clouds from observational and LES perspectives, respectively.

"The paper would benefit from improved organization and clearer, more concise writing. Specifically, Section 4 (with the note that there is no Section 5 between sections 4 and 6) is lengthy and could be subdivided into subsections. The latter part of Section 4 could potentially be formed into a subsection that addresses the limitations of the study. Additionally, the introduction section should concentrate on past studies and their relevance to the current research. Paragraphs that discuss the results and the methodology of the present study could be removed from the introduction. Furthermore, the abstract appears overly detailed. Consider revising to make it more concise while still providing essential information."

These suggestions on organization are very helpful. We agree that Section 4 could be subdivided and that results and methodology should not be featured in the introduction. The abstract was edited for brevity.

Section 4 has been subdivided into two sections:

L364: **4 Results and Discussion**

L365: **4.1 CFODD analysis to constrain ERFaci due to warm rain processes**

L486: **4.2 Limitations of CFODD-based constraint on ERFaci**

L81: Text with details about results and methodology were removed.

Abstract L19 and L30: Details were removed, and the length was decreased to adhere to 250 word limit.

"The manuscript frequently directs the reader to key figures in the supplement. It may be advantageous to consider incorporating some of these figures into the main manuscript to enhance accessibility."

We have added Figures S8-11 to the main text (now Figs. 4-5) to make the results on the PI LWP, cloud fraction and ERFaci response to autoconversion experiments more accessible.

"Including a brief description of the CFODD method would be beneficial for readers who may be less familiar with such diagrams. This addition could provide clarity and context for those who may not have an in-depth understanding of the CFODD method."

We have added the following to Section 2 to provide more details about the CFODD construction:

L129: ICOD ($\tau_d$) is parameterized as a function of MODIS COT ($\tau_c$) by invoking the adiabatic condensation growth model to vertically slice the column COT into each layer (Suzuki et al., 2010). The relationship between $\tau_d$ and $\tau_c$ is as follows:

$$\tau_d(h) = \tau_c \left\{ 1 - \left(\frac{h}{H}\right)^{5/3} \right\} \tag{1}$$

where $h$ is height and $H$ is the geometric thickness of the cloud. The detailed derivation of the ICOD coordinate is provided in Suzuki et al. (2010). The slope of the resulting 2D-PDF diagnostic is modulated by droplet collection efficiency, with steeper slope implying higher efficiency. The CFODD shows where, with ICOD on the y-axis as a vertical coordinate, the droplet collection efficiency increases, and where the transition from non-precipitating to drizzling and raining occurs, using the radar reflectivity as a proxy for the precipitation rate as described above. CFODDs are also typically binned by $R_e$ to reveal how droplet collection efficiency changes with droplet size (Suzuki et al., 2010; Takahashi et al., 2021; Jing et al., 2017).

"One of the modifications to CFODD applied by the authors is reducing the cloud optical thickness (COT) to 0.3. As the authors mention, this adjustment has a substantial impact on cloud occurrence. It is worth considering whether this modification could also influence the slope of the CFODD by altering the weight of the bins with low COT (does the fitting account for data density?)."

Yes, the fitting does account for data density and the reviewer is correct that by increasing the number of samples at low COT, the weight from thin SLWCs increases. This is consistent with the way the CFODD has been fit in prior studies (e.g., Suzuki et al., 2010) and is worth mentioning.

L143: The regression was applied to the MODIS-CloudSat profiles and E3SMv2 output at $4 \leq$ ICOD $\leq 20$ and $Z < 20$ dBZ. For E3SMv2 output, the regression was applied to approximated source CloudSat reflectivity and ICOD data that was estimated from time-mean CFODD frequencies.

L259: The decreased COT threshold also increases the weight of optically thin SLWCs, as the linear regression is applied to the CFODD source data directly (i.e., the ICOD and reflectivity profiles).

"Clarification is also needed on whether COT represents the in-cloud mean or the grid mean values. Specifying the resolution of the data is missing and it is unclear how products with different resolutions are combined."

The following text has been added to specify the resolution of the different simulated MODIS and CloudSat products in COSPv2.0:

L229: The simulated MODIS COT represents in-cloud mean, as do the other MODIS variables used in the WRDs (e.g., LWP, $R_e$). For example, the MODIS liquid COT is computed by averaging the MODIS liquid COT in cloudy subcolumns across the grid-box. In E3SMv2-COSP, the same in-cloud stratiform COT value from the E3SMv2 radiative transfer module is distributed across all the subcolumns designated as stratiform cloud by SCOPS, as described above. These values and cloud/clear-sky designations for each subcolumn are used as input to the MODIS simulator to calculate the in-cloud MODIS liquid COT. Subcolumn-level SLWC reflectivity profiles are used as input to the WRDs, also with cloud properties homogenously distributed across the subcolumns of a given classification. Thus, in E3SM-COSP, the SLWC samples within a gridbox that have the same subcolumn

classification (i.e., stratiform liquid or stratiform rain) will have the same MODIS COT and CloudSat reflectivity profile.

We have also added some details about the resolution of the MODIS and CloudSat satellite products. We assume that the reviewer is not asking for details about how the MODIS retrievals are mapped to the CloudSat footprint (1.4 x 1.8 km) in the MOD06-1KM-AUX product (1 km resolution) but we do include the Platnick et al. (2017) reference where this information is provided.

L249: The MOD06-1KM-AUX R05 product (Platnick et al., 2017), which provides MODIS collection 6 retrievals along the CloudSat footprint at 1 km resolution, supplied the 6 MODIS cloud retrievals required for the SLWC detection […]

L252: Atmospheric temperature profiles were obtained from ECMWF-AUX R05, which includes temperature profiles from the European Centre for Medium-Range Weather Forecast (ECMWF) model interpolated to the CloudSat footprint. 2B-GEOPROF R05 provided the CloudSat reflectivity profiles, the Cloud Profiling Radar (CPR) cloud mask and echo top characterization at 1.8 km resolution (Marchand et al., 2008).

"L266: There is a reference typo that needs correction."

Typo corrected (L307), thank you.

"L309: The statement about how optically thin layers can have high reflectivities near the cloud top is unclear. It would be helpful to clarify whether this refers to the small optical thickness of thin layers at the top of deeper clouds. The entire sentence needs clarification for better understanding."

L348 has been updated: "The high reflectivities near the cloud top are pronounced in the subset of E3SMv2 SLWCs with 4 < MODIS COT < 20 (Fig. S3), indicating that the high reflectivity at low ICOD in Figs. 2 (j-l) is not just a product of a subset of precipitating, optically thin SLWCs, but that layers near the cloud top in deeper SLWCs are also precipitating."

"L310: The term "strange features" needs clarification. Specify and elaborate on what exactly is being referred to."

This term has been removed from L348.

"L315: It would be beneficial to explicitly mention that you are describing Figure 2 in this section for better guidance to the reader."

L356 has been updated: "Figure 2 shows absolute frequencies of SLWCs binned by MODIS COT in each CFODD $R_e$ bin for the updated A-Train analysis (Fig. 2 g-i) and E3SMv2 only (Fig. 2 m-o)."

"L337: Clarify the meaning of "ERFaciSW strengthens." Provide a physical explanation, particularly in relation to its impact on cloud cover and albedo. This clarification will enhance the understanding of the reader."

The following has been added to L383: "As CFODD slopes represent an estimate of droplet collection efficiency, Fig. 3 demonstrates that ERFaci$_{SW}$ strengthens (increases in magnitude) with increasing droplet collection efficiency in E3SMv2 SLWCs with $R_e$ between 5 and 18 µm. One possible physical explanation for the relationship between autoconversion, droplet collection efficiency, and ERfaci$_{SW}$ is that increased autoconversion rates increase the susceptibility of clouds to precipitation suppression by aerosols. For a given optical depth, SLWCs with lower LWP and/or higher $N_d$ will precipitate more when the autoconversion rate is increased. A larger population of precipitating SLWCs results in increased susceptibility to precipitation suppression by aerosols overall. When aerosols suppress precipitation (e.g., Suzuki et al., 2013), LWP and/or cloud fraction may be enhanced, resulting in brighter clouds and stronger ERFaci$_{SW}$. The relationship between aerosols, LWP and cloud fraction (Albrecht, 1989), however, is highly uncertain, varies regionally (Sato et al., 2018), and is influenced by processes that are buffered over multiple spatiotemporal scales (Stevens and Feingold, 2009)."

L348: It is recommended to reword this sentence and provide a clear explanation that the observed slopes are utilized to constrain the simulated ERFaci. This sentence essentially describes the core of the study.

L408 has been updated: "In Figure 3, we constrain ERFaci$_{SW}$ due to autoconversion uncertainties using the linear regression between the simulated CFODD slopes and ERFaci$_{SW\_SLWCs}$. We estimated the constrained value of ERFaci$_{SW\_SLWCs}$ at the intercept of the linear relationship with the observed MODIS-CloudSat CFODD slope (Fig. 4)."

L350: Suggest marking -0.067 on the Figure for clarity and reference.

We added a marker to Fig. 3 for the constrained ERFaci$_{SW}$ value for SLWCs as suggested.

L351: Clarify the meaning of the total ERFaci of -1.5 in the context of the discussion. Mention again that it is the difference between the simulations (as stated in Line 249). Address why this value cannot be deduced directly from Figure 3.

We thank the reviewer for bringing our attention to the points that need clarification for improved readability. The point that ERFaci$_{SW}$ is calculated for SLWCs only will be made more apparent throughout the text, including for the figures. All references to ERFaci$_{SW}$ calculated for only SLWCs will be changed to ERFaci$_{SW\_SLWCs}$. We will also add some clarification to the paragraph beginning on L408 as suggested:

In Figure 3, we constrain ERFaci$_{SW}$ due to autoconversion uncertainties using the linear regression between the simulated CFODD slopes and ERFaci$_{SW\_SLWCs}$. ERFaci$_{SW}$ and ERFaci$_{SW\_SLWCs}$ values were calculated following Ghan et al. (2013), which considers the difference in TOA radiative flux between the PD and PI experiments neglecting direct forcing of aerosols (see Sect. 2.4 for details). We estimated the constrained value of ERFaci$_{SW\_SLWCs}$ at the intercept of the linear relationship with the observed MODIS-CloudSat CFODD slope (Fig. 4). The ERFaci$_{SW\_SLWCs}$ predicted by the linear regression at the MODIS-CloudSat slope value is -0.066 W m$^{-2}$, a 14 ± 6% decrease in magnitude compared to the ERFaci$_{SW\_SLWCs}$ value predicted by the E3SMv2 CNTL simulation (-0.077 W m$^{-2}$). E3SMv2's total ERFaci (-1.50 Wm$^{-2}$), inclusive of all cloud types and the longwave forcing component, falls within the IPCC AR6 'very likely' range for ERFaci (-1.0 ± 0.7 Wm$^{-2}$). The shortwave component of ERFaci is significantly larger than longwave in CMIP6 models (e.g., multimodel means of -0.91 and +0.10 W m$^{-2}$, respectively, as reported in Smith et al. 2020). Thus, our results indicate that eliminating the bias in ERFaci$_{SW}$ due to autoconversion uncertainties would decrease the magnitude of ERFaci$_{SW}$ and bring the predicted total ERFaci closer to the median IPCC ERFaci value (Forster et al., 2021).

L352: Specify whether "correcting" means changing the coefficients in the autoconversion parameterization.

The term "correcting" was removed from this line. We wouldn't want to pigeon-hole the solution into adjusting KK2000 coefficients when there are innumerable possibilities for decreasing the bias from this process representation.

L418: Thus, our results indicate that eliminated the bias in ERFaci$_{SW}$ due to autoconversion uncertainties would decrease the magnitude of ERFaci$_{SW}$ and bring the predicted total ERFaci closer to the median IPCC ERFaci value (Forster et al., 2021).

"L390: Why "however"? The linear relationship applies to both medium-large and small Re."

Thank you for catching this. There is a typo in this line that has been updated. The "however" is referring to the positive linear relationship in the small $R_e$ bin, which contrasts with medium and large $R_e$ bins:

L473: "The small $R_e$ SLWCs, however, show a positive correlation between ERFaci$_{SW}$ and CFODD slope, indicating that ERFaci$_{SW}$ weaken$_s$ as autconversion rates increase, likely due to decreased precipitation suppression susceptibility in this subset."

"L460: Elaborate on the statement that "simulated reflectivity profiles near the cloud top are decoupled from the cloud below." Provide a more detailed explanation of what is meant by "decoupled." This is the only instance where "decoupled" is mentioned in the manuscript."

L550 has been updated for clarity: "The E3SMv2 CFODD results also show reflectivities exceeding 0 dBZ near cloud top at 2 < ICOD < 4 yet relatively low reflectivities at ICOD > 5. The unreasonably high reflectivities near cloud top may indicate artifacts due to inconsistencies between E3SMv2 outputs and COSPv2.0 inputs to the CloudSat simulator. This issue motivates further investigation in future studies involving applications of the CloudSat simulator to E3SM.

References:

Ackerman, A. S., Kirkpatrick, M. P., Stevens, D. E., and Toon, O. B.: The impact of humidity above stratiform clouds on indirect aerosol climate forcing, Nature, 432, 1014-1017, 10.1038/nature03174, 2004.

Albrecht, B. A.: Aerosols, Cloud Microphysics, and Fractional Cloudiness, Science, 245, 1227--1230 , pmid = 17747885, 10.1126/science.245.4923.1227, 1989.

Danabasoglu, G., Lamarque, J.-F., Bacmeister, J., Bailey, D. A., DuVivier, A. K., Edwards, J., Emmons, L. K., Fasullo, J., Garcia, R., Gettelman, A., Hannay, C., Holland, M. M., Large, W. G., Lauritzen, P. H., Lawrence, D. M., Lenaerts, J. T. M., Lindsay, K., Lipscomb, W. H., Mills, M. J., Neale, R., Oleson, K. W., Otto-Bliesner, B., Phillips, A. S., Sacks, W., Tilmes, S., van Kampenhout, L., Vertenstein, M., Bertini, A., Dennis, J., Deser, C., Fischer, C., Fox-Kemper, B., Kay, J. E., Kinnison, D., Kushner, P. J., Larson, V. E., Long, M. C., Mickelson, S., Moore, J. K., Nienhouse, E., Polvani, L., Rasch, P. J., and Strand, W. G.: The Community Earth System Model Version 2 (CESM2), Journal of Advances in Modeling Earth Systems, 12, e2019MS001916, https://doi.org/10.1029/2019MS001916, 2020.

Gettelman, A., Hannay, C., Bacmeister, J. T., Neale, R. B., Pendergrass, A. G., Danabasoglu, G., Lamarque, J.-F., Fasullo, J. T., Bailey, D. A., Lawrence, D. M., and Mills, M. J.: High Climate Sensitivity in the Community Earth System Model Version 2 (CESM2), Geophysical Research Letters, 46, 8329-8337, https://doi.org/10.1029/2019GL083978, 2019.

Golaz, J.-C., Caldwell, P. M., Van Roekel, L. P., Petersen, M. R., Tang, Q., Wolfe, J. D., Abeshu, G., Anantharaj, V., Asay-Davis, X. S., Bader, D. C., Baldwin, S. A., Bisht, G., Bogenschutz, P. A., Branstetter, M., Brunke, M. A., Brus, S. R., Burrows, S. M., Cameron-Smith, P. J., Donahue, A. S., Deakin, M., Easter, R. C., Evans, K. J., Feng, Y., Flanner, M., Foucar, J. G., Fyke, J. G., Griffin, B. M., Hannay, C., Harrop, B. E., Hoffman, M. J., Hunke, E. C., Jacob, R. L., Jacobsen, D. W., Jeffery, N., Jones, P. W., Keen, N. D., Klein, S. A., Larson, V. E., Leung, L. R., Li, H.-Y., Lin, W., Lipscomb, W. H., Ma, P.-L., Mahajan, S., Maltrud, M. E., Mametjanov, A., McClean, J. L., McCoy, R. B., Neale, R. B., Price, S. F., Qian, Y., Rasch, P. J., Reeves Eyre, J. E. J., Riley, W. J., Ringler, T. D., Roberts, A. F., Roesler, E. L., Salinger, A. G., Shaheen, Z., Shi, X., Singh, B., Tang, J., Taylor, M. A., Thornton, P. E., Turner, A. K., Veneziani, M., Wan, H., Wang, H., Wang, S., Williams, D. N., Wolfram, P. J., Worley, P. H., Xie, S., Yang, Y., Yoon, J.-H., Zelinka, M. D., Zender, C. S., Zeng, X., Zhang, C., Zhang, K., Zhang, Y., Zheng, X., Zhou, T., and Zhu, Q.: The DOE E3SM Coupled Model Version 1: Overview and Evaluation at Standard Resolution, Journal of Advances in Modeling Earth Systems, 11, 2089-2129, https://doi.org/10.1029/2018MS001603, 2019.

Jing, X., Suzuki, K., Guo, H., Goto, D., Ogura, T., Koshiro, T., and Mülmenstädt, J.: A Multimodel Study on Warm Precipitation Biases in Global Models Compared to Satellite Observations, Journal of Geophysical Research: Atmospheres, 122, 11, 806-811, 824, https://doi.org/10.1002/2017JD027310 , issue = 21, 2017.

Mulcahy, J. P., Jones, C. G., Rumbold, S. T., Kuhlbrodt, T., Dittus, A. J., Blockley, E. W., Yool, A., Walton, J., Hardacre, C., Andrews, T., Bodas-Salcedo, A., Stringer, M., de Mora, L., Harris, P., Hill, R., Kelley, D., Robertson, E., and Tang, Y.: UKESM1.1: development and evaluation of an updated configuration of the UK Earth System Model, Geosci. Model Dev., 16, 1569-1600, 10.5194/gmd-16-1569-2023, 2023.

Mülmenstädt, J. and Feingold, G.: The Radiative Forcing of Aerosol–Cloud Interactions in Liquid Clouds: Wrestling and Embracing Uncertainty, Current Climate Change Reports, 4, 23-40, 10.1007/s40641-018-0089-y, 2018.

Sato, Y., Goto, D., Michibata, T., Suzuki, K., Takemura, T., Tomita, H., and Nakajima, T.: Aerosol effects on cloud water amounts were successfully simulated by a global cloud-system resolving model, Nature Communications, 9, 985, 10.1038/s41467-018-03379-6, 2018.

Smith, C. J., Kramer, R. J., Myhre, G., Alterskjr, K., Collins, W., Sima, A., Boucher, O., Dufresne, J.-L., Nabat, P., Michou, M., Yukimoto, S., Cole, J., Paynter, D., Shiogama, H., O'Connor, F. M., Robertson, E., Wiltshire, A., Andrews, T., Hannay, C., Miller, R., Nazarenko, L., Kirkevg, A., and Olivi: Effective radiative forcing and

adjustments in CMIP6 models, Atmospheric Chemistry and Physics, 20, 9591--9618, 10.5194/acp-20-9591-2020, 2020.

Stevens, B. and Feingold, G.: Untangling aerosol effects on clouds and precipitation in a buffered system, Nature, 461, 607-613, 10.1038/nature08281, 2009.

Suzuki, K., Nakajima, T. Y., and Stephens, G. L.: Particle Growth and Drop Collection Efficiency of Warm Clouds as Inferred from Joint CloudSat and MODIS Observations, Journal of the Atmospheric Sciences, 67, 3019-3032, 10.1175/2010JAS3463.1, 2010.

Takahashi, H., Bodas-Salcedo, A., and Stephens, G.: Warm Cloud Evolution, Precipitation, and Their Weak Linkage in HadGEM3: New Process-Level Diagnostics Using A-Train Observations, Journal of the Atmospheric Sciences, 78, 2075-2087, https://doi.org/10.1175/JAS-D-20-0321.1, 2021.

Zhang, Y., Xie, S., Qin, Y., Lin, W., Golaz, J. C., Zheng, X., Ma, P. L., Qian, Y., Tang, Q., Terai, C. R., and Zhang, M.: Understanding changes in cloud simulations from E3SM version 1 to version 2, Geosci. Model Dev., 17, 169-189, 10.5194/gmd-17-169-2024, 2024.

---

## Author Response (AR2)

Response to Report #1:

We thank the reviewer for their attention to the revised manuscript and the thoughtful discussion. Our responses and line-by-line changes are below.

"It seems that the authors are putting far to much credulity in the linear regression relating the CFODD slopes to ERF_ACI, when the CNTRL simulation already almost perfectly matches the observation. I think this manuscript is borrowing way too heavily from the methodology used in the 'emergent constraints' world, where in many studies the control variable is very different between the simulation and the observation and it might make sense to use the linear regression to get a good estimate of ECS. If you really want to use this linear regression as quantitatively as you have you should prove that it is robust. For example, do several more experiments where you vary other model parameters and see how that changes your slope. I can almost guarantee that you can change it outside of the bounds of your existing uncertainty range.

My strong recommendation is to remove the repeated mentioning of reduced ERF_ACI by constraining the CFODD shape by such a precise number and instead just state the model is in agreement with observations and there is a strong sensitivity of ERF_ACI to this particular metric which provides a useful constraint on a highly uncertain process."

We agree that a different result and constrained value of $ERFaci_{SW}$ would be expected if we repeated the experiment for other model parameters that modulate droplet collection efficiency. This is stated on L419 and L529:

L419: As $ERFaci_{SW}$ is the result of many cloud processes, the updated CFODD analysis should be interpreted as a constraint on the component of $ERFaci_{SW}$ that is modulated by droplet collection efficiency due to autoconversion. In other words, the updated CFODD analysis shows the change in $ERFaci_{SW}$ one would expect if the bias in $ERFaci_{SW}$ due to a specific process representation affecting droplet collection efficiency were eliminated.

L529: While this study focuses on autoconversion, future studies should apply CFODD analysis to other microphysical processes that affect droplet collection efficiency (e.g., accretion, droplet breakup, evaporation) to generate additional ERFaci constraints.

In other words, we would expect a different result for another process. The CFODD analysis shows you what change in $ERFaci_{SW}$ you could expect if the bias in $ERFaci_{SW}$ due to a given process were reduced.

We also recognize that there are additional uncertainties in the linear regression approach to constraining ERFaci due to limitations of our study that are not represented in the 95% confidence interval, such as the limited number of experiments. We have modified the text as the reviewer suggested and have removed all statements referring to reducing $ERFaci_{SW}$ by a specific quantity.

Abstract: E3SMv2's CFODD slope (0.20 ± 0.04) is in agreement with observations (0.20 ± 0.03). The strong sensitivity of ERFaci$_{SW}$ to the CFODD slope provides a useful constraint on highly uncertain warm rain processes, whereby ERFaci$_{SW}$, constrained by MODIS-CloudSat, is estimated by calculating the intercept of the linear association between the ERFaci$_{SW}$ and the CFODD slopes, using the MODIS-CloudSat CFODD slope as a reference.

L400: The constrained value of ERFaci$_{SW\_SLWCs}$ is estimated at the intercept of the linear relationship with the observed MODIS-CloudSat CFODD slope (Fig. 4). We find that the ERFaci$_{SW\_SLWCs}$ predicted by the linear regression at the MODIS-CloudSat slope value (-0.066 W m$^{-2}$ ± 0.06 W m$^{-2}$) approaches agreement with the ERFaci$_{SW\_SLWCs}$ value predicted by the E3SMv2 CNTL simulation (-0.077 W m$^{-2}$), particularly considering the additional uncertainties imposed by the limited number of sensitivity experiments that are not represented in the regression's 95% confidence interval. The agreement between the constrained and predicted value of ERFaci$_{SW\_SLWCs}$ indicates that the ERFaci$_{SW}$ due to autoconversion is well-represented in E3SMv2 according to CFODD analysis.

L521: In this study, we present an updated CFODD analysis, demonstrate how it can be applied to ESMs as a process-oriented constraint on ERFaci and find that E3SMv2's ERFaci$_{SW}$ agrees with the MODIS-CloudSat constrained value within uncertainty.

We have also updated L472 because important context was removed in the preceding paragraph on results due to the updates above:

L472: Considering that constrained ERFaci$_{SW}$ increases in magnitude with increasing R$_e$ in Fig. S7,that the shortwave component of ERFaci is significantly larger than the longwave in CMIP6 models (Smith et al., 2020), and that E3SMv2's total ERFaci (-1.50 W m$^{-2}$) is relatively strong compared to the IPCC AR6 'very likely' range (-1.0 ± 0.7 W m$^{-2}$) (Forster et al., 2021), the underrepresentation of SLWCs with large R$_e$ in E3SMv2 represents a compensating bias, without which the total ERFaci would be even stronger compared to IPCC AR6.